# Perspectives of HPV vaccine decision-making among young adults: A qualitative systematic review and evidence synthesis

Namoonga M. Mantina[1]*, Jonathan Smith[1], Flavia Nakayima Miiro[2], Priscilla Anne Magrath[1], Deborah Jean McClelland[1,3], Leila Barraza[4], John Ruiz[5], Purnima Madhivanan[1,6]

**1** Department of Health Promotion Sciences, Mel & Enid Zuckerman College of Public Health, University of Arizona, Tucson, Arizona, United States of America, **2** Department of Epidemiology, Mel & Enid Zuckerman College of Public Health, University of Arizona, Tucson, Arizona, United States of America, **3** Arizona Health Sciences Library, University of Arizona, Tucson, Arizona, United States of America, **4** Department of Public Health Practice, Policy, & Translational Research, Mel & Enid Zuckerman College of Public Health, University of Arizona, Tucson, Arizona, United States of America, **5** Department of Psychology, College of Sciences, University of Arizona, Tucson, Arizona, United States of America, **6** Division of Infectious Diseases, Department of Medicine, College of Medicine, University of Arizona, Tucson, Arizona, United States of America

* mantinan@arizona.edu

## Abstract

### Background

Despite the demonstrated safety and effectiveness of HPV vaccines in preventing HPV-related cancers, global vaccine coverage remains low. The suboptimal adolescent HPV vaccine coverage rate leaves many young adults at increased risk for developing vaccine preventable HPV-related cancers. This qualitative evidence synthesis (QES) aims to examine the HPV vaccination perspectives of young adults globally and identify the barriers and facilitators to HPV vaccine uptake and decision-making processes.

### Methods

A comprehensive search was conducted on October 2023 across seven databases to identify studies that reported on HPV vaccination among young adults aged 18–26 years and used qualitative study methods or analysis techniques.

### Results

Forty-two studies were purposively sampled for inclusion, presenting 29 findings across 10 thematic categories. Vaccine eligible young adults believed that they had aged out of eligibility for HPV vaccination. There was also a perspective that condom use, and regular screenings were alternatives to vaccination in preventing HPV infections. Challenges included scheduling appointments, requirements for multiple

**Data availability statement:** The data reported in this manuscript are publicly available through the University of Arizona Research Data Repository (ReDATA) at the following DOI: https://doi.org/10.25422/azu.data.28355615

**Funding:** The author(s) received no specific funding for this work.

**Competing interests:** The authors have declared that no competing interests exist.

shots, and vaccine cost. There was also concern for the gendered nature of vaccine promotion. Lastly, despite being at the age to make autonomous decisions, parents were still influential and active in the vaccine decision-making process for their children.

## Conclusion

The novelty of this study, as one of the principal QES on catch-up HPV vaccination, presents findings that underscore the complexity of factors across multiple ecological levels which may aid or impede vaccination uptake among young adults and provide important considerations for interventions, programs, and policies aimed at addressing HPV vaccination disparities among young adults.

## Background

Cancer is among the top two leading causes of death across 127 countries globally [1]. The human papillomavirus (HPV) is responsible for approximately 4.5% of all cancer cases worldwide [2] and persistent HPV infection can lead to cancers of the cervix, penis, vulva, vagina, anus, and oropharynx [3]. Cervical cancer is the second most diagnosed cancer among women and second leading cause of death among women worldwide [4]; this has significant societal implications, for example, nearly 210,000 children globally become orphans because their mother died from cervical cancer in 2020 [5].

The first HPV vaccine became publicly available in 2006. The Centers for Disease Control and Prevention (CDC) advocate for the initiation of a two-dose HPV vaccination series in adolescents aged 9–14 and a three-dose regimen for teenagers and young adults between ages 15 and 26 [6]. In December 2022, the World Health Organization (WHO) updated its guidance on the HPV vaccine, introducing a single-dose scheduling option for girls and women aged 9–20 years [7,8]. Despite the demonstrated safety and effectiveness of HPV vaccines in preventing HPV-related cancers, global adolescent vaccine coverage continues to remain low, estimated at only 15% for females and 4% for males [9]. HPV vaccination during early adolescence is critical in reducing the risk of HPV-related cancers. However, not all adolescents receive the vaccine, leading to disparities in vaccination rates across regions [9,10]. Additionally, individuals aged 15–24 account for nearly half of all new sexually transmitted infections (STI) cases annually [11] and it's estimated that over half of HPV infections that develop into cancer are acquired by the age of 21 [12]. The suboptimal adolescent coverage rate leaves many young adults at increased risk for developing vaccine preventable HPV-related cancers; these young adults remain eligible for the vaccine and have increased autonomy in making healthcare decisions and shaping their health behaviors [13].

Young or emerging adulthood is a distinct developmental phase from adolescence [14] that is characterized by identity exploration, instability, and feeling "in-between", which result from pivotal life transitions, a focus on "self" and

optimism of the possibilities available [15]. Effectively promoting HPV vaccination among young adults requires a deeper comprehension of the unique factors characterizing the vaccination behaviors and influences of this demographic. Interventions to increase vaccination uptake in young adults have had varying effectiveness; prior reviews have reported that majority of interventions designed to increase HPV vaccination in young adults did not result in significant increases in vaccine uptake [16,17]. Interventions also lack participant diversity [18,19] which further exacerbates HPV-related disease inequities and undermines the discovery of how different intervention components may work for diverse populations [19,20].

Although several systematic reviews have evaluated HPV vaccination interventions among young adults, there remains a need for further research in this area to enhance our understanding of HPV vaccine promotion and decision-making for these individuals. One review examining interventions to promote HPV vaccination globally discovered that only 25% of interventions conducted between 2015 and 2020 targeted young adults aged 18–34 years [21]. Furthermore, the broad age range of participants included in reviews presents challenges in understanding the unique factors faced by young adults aged 18–26 years who are eligible for HPV vaccination. For instance, two reviews assessing the effectiveness of HPV interventions targeted individuals 9–26 years [18,22] and another review identifying effective strategies to enhance HPV vaccine uptake encompassed participants aged 11–26 years [19]. Notably, nearly all the systematic review literature on HPV vaccination among young adults have been quantitative syntheses. Qualitative research is suitable for exploring the intricate nuances and complexities that shape vaccination behavior and decision-making processes, subsequently providing a better understanding of how various factors interplay. The only identified qualitative systematic review on HPV vaccination did not have a registered review protocol and exclusively explored adolescent girls aged 9–18 years in high income countries [23].

Given these collective factors—the suboptimal adolescent HPV vaccine coverage rate, limited HPV vaccine interventions targeting young adults, and the wide age range of young adult participants included in HPV vaccine research—there is a crucial opportunity to specifically intervene and promote catch-up HPV vaccination among young adults aged 18–26. To the extent known, there is no published systematic qualitative evidence synthesis of global literature on HPV vaccination among young adults 18–26 years old. The aim of this qualitative evidence synthesis (QES) is to examine the HPV vaccination perspectives of young adults aged 18–26 years and identify barriers and facilitators to HPV vaccine uptake and decision-making processes for this population.

## Methods

This systematic review adheres to the reporting guidelines of the Preferred Reporting Items for Systematic Review and Meta-Analysis (PRISMA) [24] and the QES guidelines detailed by Cochrane [25]. This review was conducted in accordance with the review protocol that was registered with the International Prospective Register of Systematic Reviews (PROSPERO) on April 30, 2023 (PROSPERO registration number CRD42023417052) and published with BMJ Open [26]. Any amendments or deviations from the protocol are documented and reported in this manuscript.

### Review team reflexivity

This systematic review engaged a multidisciplinary team of researchers who represented diverse social identities and research experiences pertinent to the topic. Academic disciplines represented include health promotion sciences, epidemiology, and medical anthropology. Full details of the review team are documented in the protocol paper for this manuscript [26].

### Study eligibility criteria

This review evaluated primary studies that reported on the perspectives, factors and/or decision-making processes of young adults aged 18–26 years. The inclusion and exclusion criteria for this review as aligned with the protocol follow below.

**Inclusion criteria.**

• Reported on HPV vaccination views, perspectives, decision of individuals aged 18–26 years,

• Studies that used qualitative study designs (e.g., ethnography, case studies),

• Studies that used qualitative methods for data collection (e.g., focus groups, interviews, open-ended survey questions),

• Studies that used qualitative data analysis methods (e.g., thematic analysis, grounded theory)

• Mixed methods studies where it is possible to extract data that were collected and analyzed using qualitative methods, and

• We included studies regardless of their linkage to an intervention and irrespective of where vaccination took place or how it was delivered.

**Exclusion criteria.**

• Studies that did not include young adults 18–26 years old and solely presented the perspectives and views of other stakeholders (such as parents, healthcare providers, and policymakers) on the topic of young adults HPV vaccination uptake,

• Articles that were protocols, reviews, conference proceedings and opinion reports,

• Studies that collected data using qualitative methods but did not analyze these data using qualitative analysis methods (e.g., open-ended survey questions where the response data are analyzed using descriptive statistics only),

• Studies that evaluated multiple vaccines, but did not report on HPV separately or it was not possible to extract data specific to HPV,

• Studies where it was not possible to separate and extract data on views of HPV vaccination specific to young adults from views of vaccination of other age groups/demographics (e.g., adolescents under 18 years, studies that included a wide age range of participants, but the qualitative data reported did not explicitly state the age of the participant quoted).

## Information sources and search strategy

We searched the following seven (7) electronic databases to identify studies for inclusion: PubMed, SCOPUS (Elsevier), Embase (Elsevier), the Cochrane Library (Wiley), PsycINFO (EBSCOhost), CINAHL (EBSCOhost) and CABI Global Health (EBSCOhost). No search restrictions were posed on geographic region/country where studies were conducted, nor on language of the manuscript. A comprehensive search strategy was developed and adapted for each database. Key words and Medical Subject Headings (MeSH) terms were combined related to "HPV", "vaccination", "young adults" and qualitative methodological filters (e.g., "focus group", "interview"). Full details of the search terms for each database are details in S1 Appendix.

All databases were searched by NMM with support from DJM beginning in 2006 (the year HPV vaccines were publicly available) up to the date of the search. The initial search was conducted on May 11, 2023. Updates were made to the search terms following peer review feedback on the protocol manuscript and the search was rerun on October 12, 2023. As screening procedures had commenced following the initial search, we followed the procedures outlined by Bramer & Bain [27] to identify the new records retrieved from the updated search (S2 Appendix). Lastly, the reference lists of all eligible studies were also manually searched to identify additional studies for inclusion.

## Selection of studies

The citation manager software EndNote (Clarivate, London, UK) was used to manage and deduplicate all the studies identified through the electronic database search. The deduplicated citation file was then imported into the literature

review software DistillerSR (Evidence Partners, Ontario, Canada) for screening in accordance with the review protocol. DistillerSR further identified additional duplicate records which were removed using the software's functionality.

Studies were screened through a title/abstract review stage and a full text review stage to ascertain inclusion as guided by the eligibility criteria. Each study was independently assessed by two review authors at each screening stage. Any disagreements in evaluation were resolved by discussion and consensus between reviewers. While there were no language restrictions implemented during the database search, there were two non-English full text articles obtained (one Korean and one Spanish). To facilitate the review for inclusion, Google Translate was initially used to aid with reviewing the methods section of the articles. It was determined that both articles did not meet inclusion criteria at full text and were subsequently excluded. After the two stages of screening, 71 articles met the review eligibility criteria for consideration to be included in the review.

## Sampling of studies

In QES, managing the volume of data examined is crucial to uphold analysis quality [28]. Maximizing the heterogeneity of identified themes and concepts to foster a more profound understanding of the research subject is prioritized over exhaustively assessing every study [29]. Consequently, purposively sampling methods were used to aid the management and analysis of the qualitative data to be extracted. In alignment with the approaches taken by other qualitative review authors [30], we purposively sampled a subset of the studies for analysis, using an approach called criterion sampling [31,32], in which studies are sampled based on meeting predetermined criteria. Given the aim of the review was to understand HPV catchup vaccination in individuals aged 18–26, the age criterion employed for sampling inclusion was that studies exclusively recruited participants 18–26 years old. We believed that this sampling criterion would best facilitate obtaining focused and rich information essential to the research question. Forty-five (45) articles were included in the review following purposive sampling.

## Data Extraction

The data extraction process was performed in 2 steps, an adjustment from the original protocol. Step 1 was the extraction of the study characteristics, and step 2 was the extraction of qualitative data. Aligned with the protocol, step 1 of data extraction was performed in DistillerSR, using a data extraction form specifically created for this review. Two reviewers independently extracted data from each study and discrepancies were resolved by a third reviewer. The study characteristics data extracted from eligible studies at step 1 were:

- publication year

- data collection year

- context (study country; rural vs urban)

- participants (number of participants; demographic characteristics, e.g., age, gender, race/ethnicity, sexual orientation)

- study design (objective/research question, sample recruitment, data collection methods, theory frameworks or conceptual models used, analysis methods, funding)

All articles that met review eligibility (n=71) had their data extracted at step 1. This was done to be transparent about the existing literature that was applicable to research question, even though we may not report on all of it for this review.

Step 2 of the data extraction process was conducted in REDCap (Research Electronic Data Capture) [33] as the review team found it was better suited to handle qualitative data. This step was performed only on the included articles that met the criterion sampling (n=45). The information extracted in step 2 were 1) the themes or constructs developed by the primary study authors, including any contextual explanation for how the themes/concepts were defined by the primary authors; and 2) verbatim participant quotes provided. For consistency in how the qualitative data were extracted across all the articles, step 2 data extraction was performed by one review author (NMM).

## Assessment of the methodological limitations of included studies

The methodological limitations of each included study was independently assessed by two review authors using the Critical Appraisal Skills Programme (CASP) quality assessment tool for qualitative studies [34]. An adapted version of the tool was used as guided by other reviews [28,35,36] and was piloted on a subset of studies to assess feasibility and integrity of the assessment. Disagreements were resolved by a third review author.

## Data synthesis

We used Braun and Clark's methodological process of thematic analysis to analyze and synthesize the qualitative data [37]. Data was analyzed using qualitative analysis software Dedoose version 9.2.005. The following stages of thematic analysis were undertaken:

- *Familiarization*: Three review authors (NMM, JS, FNM) reviewed all the extracted data. Notes were taken on the impression of the data.

- *Generating initial codes*: The aforementioned review authors initially coded data from three different studies that had the largest amount of data extracted, to inductively identify and generate initial codes; codes were also formed from the notes taken during the familiarization stage. After this initial coding, the review authors met to discuss the codes identified, consolidate any duplicate codes, and finalize the codebook. Another round of coding was performed on three additional studies to verify the codebook; no new codes emerged. Each study was then independently coded by two review authors. During the coding process, authors were able to take additional notes in Dedoose pertaining to trends and observations they noticed as they coded. After coding, the two review authors assigned to each study met to discuss any discrepancies in coding and decide the finalized version of the coded data.

- *Generating, reviewing, and finalizing themes*: Once all the data were coded, review author NMM generated the initial themes and subthemes from the codes, and any notes take from the coding process. The initial themes generated formed the basis of the review findings which were subsequently reviewed and revised by the other authors involved in the coding process (JS, FNM).

## Assessment of confidence in synthesized findings

As recommended by the Cochrane handbook on QES, we used the GRADE- CERQual (Grading of Recommendations Assessment, Development and Evaluation - Confidence in the Evidence from Reviews of Qualitative research) [38] to evaluate the confidence in synthesized findings. The CERQual confidence assessment is based on the following four key components:

- *Methodological limitations of included studies*: the extent to which there were concerns about the design or conduct of the primary studies that contributed evidence to an individual review finding [39].

- *Coherence of the review finding*: an assessment of how clear and cogent (i.e., well-supported or compelling) the fit was between the data from the primary studies and a review finding that synthesized those data [40]

- *Adequacy of the data contributing to a review finding*: an overall determination of the degree of richness and quantity of data supporting a review finding [41].

- *Relevance of the included studies to the review question*: the extent to which the body of evidence from the primary studies supporting a review finding was applicable to the context (perspective or population, phenomenon of interest, setting) specified in the review question [42].

To facilitate this process, we used the GRADE-CERQual Interactive Summary of Qualitative Findings [43]. It is noteworthy that the evaluation of methodological limitations of the included studies was not used to exclude studies from the

review, but we did consider them in the assessment of confidence for the review findings. After evaluating each of the four components, we made a judgement on the overall confidence in each review finding; all findings started at 'high confidence' and then were graded down to 'medium,', 'low' or 'very low' confidence based on any concerns arising from the CERQual components [44].

## Results

### Results of the search and screening

We identified 1163 titles from the search of seven databases and considered 208 articles for full text review. We identified 71 articles representing 68 studies that met the review eligibility criteria. After applying purposive sampling, 45 articles representing 42 studies were included and synthesized in the review. There were 3 pairs of eligible articles that reported results from the same primary study; all three pairs were purposively sampled. The results of the search, screening, and sampling process are detailed in the PRISMA flow diagram (Fig 1).

### Description of eligible studies

This section details the study characteristics for all 68 eligible studies that met review eligibility criteria to help characterize the state of the existing qualitative literature related to HPV vaccination among young adults. Table 1 details the study characteristics of each eligible study.

**Study design and data collection.** Most studies (n=56) were qualitative, utilizing focus groups and/or interview data collection methods; one study was an autoethnography. Twelve (12) studies were mixed methods design, utilizing questionnaires in combination with interviews and/or focus groups; one study utilized retrospective chart review and interviews. Data collection was conducted between 2006 and 2021.

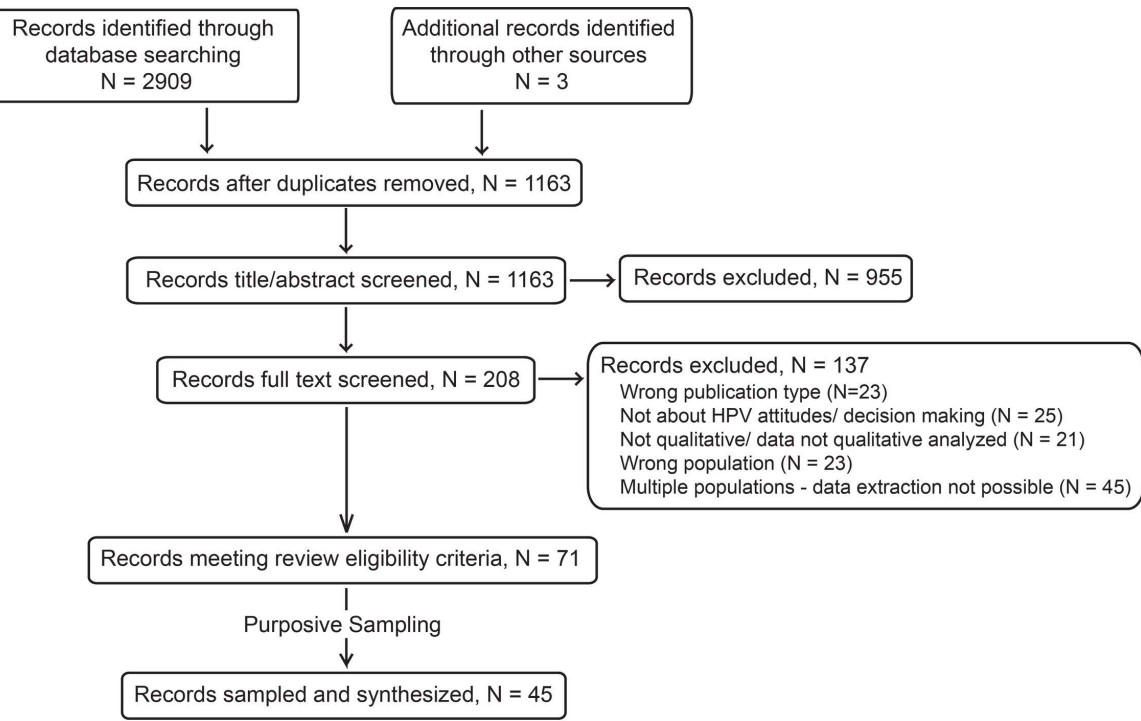

**Fig 1. Flow diagram of study selection procedures.**

PLOS One | https://doi.org/10.1371/journal.pone.0321448   May 5, 2025

**Table 1. Summary of study characteristics of eligible studies.**

| Author & Year | Study Objective | Country | Study Setting | Design & Data Collection | Theoretical Framework | Eligible Participants | Sample Size | Purposive Sampling |
|---|---|---|---|---|---|---|---|---|
| Allen 2009 [45] | "The purpose of this study was to gain a better understanding of men's knowledge and attitudes toward the HPV vaccine, to inform the development of effective population-based intervention strategies." | United States | School | Qualitative, Focus Groups | None Indicated | Students 18–22 years of age (n=45) | 45 | Yes |
| Al-Naggar 2010 [46] | "The objective of this study is to explore the perceptions and opinions of young women about human papilloma virus (HPV) vaccination and associated barriers" | Malaysia | School | Qualitative, Interviews | None Indicated | female students from different faculties [departments] | 30 | Yes |
| Apaydin 2018 [47] | "The aim of this study was to qualitatively identify patient-, provider-, and systems-level barriers and facilitators for HPV vaccination among sexual and gender minority (SGM) people." | United States | Community | Mixed Methods, Focus Groups and questionnaire | None Indicated | Sexual and gender minority (SGM) people | 15 | Yes |
| Basnyat 2018 [48] | "This study examined the means by which young Singaporean women seek and process information about HPV vaccination in their decision to become vaccinated." | Singapore | Not Specified/Uncertain | Qualitative, Interviews | Elaboration Likelihood Model | Singaporean females aged 18–26 years, having taken at least one shot of any of the two approved HPV vaccines in Singapore at any time in their lives | 26 | Yes |
| Bunton 2013 [49] | "This article reports on the first generation of young Australian women to participate in the mass HPV vaccination programme; what they knew about vaccination, why they participated, and how they reflect upon their experience" | Australia | Not Specified/Uncertain | Qualitative, Focus Groups | Grounded Theory | "1) Women aged 35 years and older, whose experience solely involved Pap smears 2) Women aged 18 years who had completed the final year of their secondary education in 2008 and been offered the HPV vaccine as part of the school based catch-up programme; and 3) Women aged 20-26 who were eligible for free vaccination under the community-based catchup programme" | 46 | No |
| Carnegie 2017 [50] | "To examine cultural barriers and participant solutions regarding acceptance and uptake of the human papillomavirus (HPV) vaccine from the perspective of Black African, White-Caribbean, Arab, Indian, Bangladeshi and Pakistani young people." | United Kingdom | Not Specified/Uncertain | Qualitative, focus groups and interviews | Discursive Psychology | Aged 16–26 years | 40 | No |
| Chan 2011 [51] | "To identify the perception on human papillomavirus (HPV) vaccination among female nursing students in Hong Kong" | China | School | Qualitative, focus groups and interviews | None Indicated | Local female nursing students were recruited in a university in Hong Kong. | 28 | Yes |

| Author & Year | Study Objective | Country | Study Setting | Design & Data Collection | Theoretical Framework | Eligible Participants | Sample Size | Purposive Sampling |
|---|---|---|---|---|---|---|---|---|
| Chen 2021 [52] | "This qualitative research explored Chinese college students' HPV-related awareness, knowledge, attitudes and beliefs, and their vaccination intention as well as the strategies promoting vaccination in China." | China | School | Qualitative, Focus Groups | Health Belief Model | An individual who was (1) a student at a university in China; (2) aged 18 and older; and (3) able to read, write, and speak Chinese. | 38 | Yes |
| Clevenger 2012 [53] | "The study's primary objective was to examine access to and knowledge of the HPV vaccine among Latina university students in Denver, Colorado." | United States | School | Qualitative, Interviews | None Indicated | Students between 19 and 26 years of age who self-identified as Latina were recruited to participate in interviews | 15 | Yes |
| Cohen 2013 [54] | "To examine differences in knowledge, attitudes, and related practices among adopters and nonadopters of the human papillomavirus (HPV) vaccine" | United States | School | Qualitative, Interviews | Diffusion of Innovation | Participants spoke English, were enrolled as a student from a large state university and were at least 18 years old but not more than 26 years old. | 83 | Yes |
| Dai 2020 [55] | "In this piece, [the author] presents a personal intercultural medical story about her HPV vaccine process. She utilizes personal narratives to share with readers her experience with sex education and to convey people's attitudes toward sexuality and sex education in mainland China." | United States | Not Specified/ Uncertain | Qualitative, Autoethnography | None Indicated | A woman born and raised in China, had no knowledge of HPV or the HPV vaccine until she came to the United States for graduate school | 1 | Yes |
| DeLauer 2020 [56] | "To identify knowledge and beliefs about the human papillomavirus (HPV) among students in a residential academic institution, including perceptions of safety of the HPV vaccine, perceptions of cancer correlation with HPV, and independence/interdependence in health decision-making." | United States | School | Qualitative, Interviews | None Indicated | | 52 | Unclear/Not Specified |
| Fields 2022 [57] | "The current study examined individual and structural motivators and barriers to HPV vaccination among medically underserved women utilizing a Planned Parenthood health center in Southeast Pennsylvania" | United States | Community | Qualitative, Interviews | Narrative Engagement Theory | Young adult (18–34) women attending a PPSP health center located in a suburb of Philadelphia, PA, USA, in the summer months between 18 June to 16 July 2015. | 24 | No |
| Fontenot 2016 [58] | "The purpose of this study was to elicit YMSM's beliefs about HPV and the HPV vaccine as well as describe perceived barriers and facilitators of vaccine initiation and completion" | United States | Not Specified/ Uncertain | Qualitative, focus groups and interviews | None Indicated | Young MSM ages 18–26 years who were able to read and understand English | 34 | Yes |

*(Continued)*

| Author & Year | Study Objective | Country | Study Setting | Design & Data Collection | Theoretical Framework | Eligible Participants | Sample Size | Purposive Sampling |
|---|---|---|---|---|---|---|---|---|
| Garcia 2023 [59] | "The objective was to investigate what HPV messages unvaccinated MA patients receive and relay to others at the individual, interpersonal, and community levels; how they ascribe meaning to these messages; and how these messages affected their intentions to vaccinate" | United States | Not Specified/ Uncertain | Qualitative, Interviews | NIHMD Theoretical Framework | Female, aged 18–26 at the time of the interview, self-identify as MA, lived in California for the past 3 years, offered the HPV vaccine by a provider in adulthood (18 years or older), and not yet initiated the HPV vaccine | 30 | Yes |
| Gerend 2019 [60] | "The purpose of this study was to identify young sexual minority men's perspectives on HPV vaccination." | United States | Community | Qualitative, Interviews | Information, Motivation, and Behavioral Skills Model (IMB) | Eligibility criteria were assigned male sex at birth; male gender identity; ages 18–26 years; self-identify as gay, bisexual, or queer; and currently live in the Chicago metro area. | 29 | Yes |
| Glenn 2021 [61] | "The purpose of this study therefore was to explore factors influencing HPV vaccine decision-making and vaccine receipt among college students in the catch-up age range, drawing on perspectives from students, healthcare providers, and staff at a large student health center" | United States | School | Qualitative, focus groups and interviews | Deductive Approach | Students between the ages of 18 and 26 years and receiving care at the student health center | 51 | No |
| Gray Brunton 2014 [62] | "Our aim was to explore meanings and/or vaccination experiences of the HPV vaccine amongst young women aged 18–26 years in four European countries with different vaccine implementation programmes: Scotland, Spain, Serbia and Bulgaria" | Multiple Countries (Bulgaria, Scotland, Serbia, Spain) | Not Specified/ Uncertain | Qualitative, Focus Groups | None Indicated | Young women were between 18 and 26 years old. | 54 | Yes |
| Head 2012 [63] | "For the present study, we sought to understand young adult women's perceptions of HPV, cervical cancer, the HPV vaccine, and Pap testing by conducting formative research to support a targeted health communication intervention to increase HPV vaccine uptake and regular Pap testing among young women in rural Appalachian Kentucky, with a special focus on the eight-county KR-ADD" | United States | Not Specified/ Uncertain | Qualitative, Interviews | Integrated Behavioral Model (IBM) | Eligible participants were women between the ages of 18 and 26 years who had received or had decided not to receive the HPV vaccine, and currently or recently resided in one of 29 eastern Kentucky counties or received medical care from the large KR-ADD medical clinic (serving the 29 counties) | 19 | Yes |
| Hirth 2018 [64] | "The purpose of this study was to evaluate motivations and barriers among community college students 18?26 years of age." | United States | School | Qualitative, Interviews | Theory of Planned Behavior | Students 18–26 years old that were currently enrolled in the community college | 19 | Yes |

*(Continued)*

| Author & Year | Study Objective | Country | Study Setting | Design & Data Collection | Theoretical Framework | Eligible Participants | Sample Size | Purposive Sampling |
|---|---|---|---|---|---|---|---|---|
| Hodge 2011 [65] | "This article reports on American Indian(AI) university students' HPV vaccine readiness and female vaccine decision-making." | United States | School | Qualitative, Focus Groups | None Indicated | Eligibility criteria included: 1) self-identified as AI; 2) aged 18–26;and 3) currently enrolled college student. | 57 | Yes |
| Hodge 2014 [66] | "This paper reports on a prelim-inary exploration of HPV among an important and often overlooked at risk subgroup, American Indian males, and the unique issues and barriers they experience, as com-pared to female participants" | United States | School | Qualitative, Focus Groups | None Indicated | Inclusion criteria included age (> 18 years), enrolled college student status, and self-identification as American Indian | 57 | Yes |
| Hopfer 2011 [67] | "The goal of this study was to better understand college women's HPV vaccine attitudes and beliefs with a lens focused on (a) the fam-ily, peer, and health care provider messages that college women report receiving about HPV, and (b) how college women interpret, respond to, and incorporate these messages to shape their own HPV vaccine attitudes and decisions" | United States | School | Qualitative, Interviews | Culture-centric Narrative | College women | 36 | Yes |
| Hopfer 2017 [68] | "The main purpose of this study was to elicit HPV vaccine decision narratives to understand how cultural values and implicit as well as explicit attitudes shape vaccine decision making among Latina and Vietnamese women attend-ing Planned Parenthood health centers" | United States | Not Specified/Uncertain | Qualitative, Interviews | Narrative Communica-tion Theory | Women ages 18–26 years, Viet-namese, or Latina | 50 | No |
| Jaiswal 2020 [69] | "The objective of this study was to elucidate the nature and depth of (a) HPV and HPV vaccine knowledge and (b) provider com-munication about HPV vaccine, in a diverse sample of young urban sexual minority men." | United States | Community | Qualitative, Interviews | None Indicated | "The parent study recruited individuals who, at baseline of Wave 2, were 22-23 years old, were assigned male at birth, had sex with a man in the previous 6 months, reported a negative or unknown HIV serostatus, and lived in the New York City metro-politan region." | 38 | Yes |
| Jin 2023 [70] | "This study examined factors associated with HPV vaccination among college students in the Mid-South of the U.S. using a mixed-methods approach." | United States | Community | Mixed Methods, interview and questionnaire | None Indicated | "Students who were aged 18 to 26 years and registered for an under-graduate program at the university at the time of the survey. Those eligible for the interviews were the participants who had completed the survey and were not up to date with the vaccination series" | 417 | Yes |

*(Continued)*

| Author & Year | Study Objective | Country | Study Setting | Design & Data Collection | Theoretical Framework | Eligible Participants | Sample Size | Purposive Sampling |
|---|---|---|---|---|---|---|---|---|
| Joseph 2014 [71] | "The purpose of this study was to examine how the knowledge, attitudes, and beliefs related to HPV disease and HPV vaccination affect vaccine uptake among young adult African-American, Haitian, Latina, and White women aged 18." | United States | Clinical | Mixed Methods, interview and questionnaire | Grounded Theory | Ages 18–22 and self-identified as White, African-American, Latina or Haitian (U.S.-born or immigrants); excluded if they were pregnant or had received HPV vaccination | 132 | Yes |
| Kim 2017 [72] | "This study aimed to explore Koran American Female students' awareness of and attitudes toward HPV vaccination" | United States | School | Qualitative, Focus Groups | Theory of Planned Behavior | Korean undergraduate students or female graduate students aged 18–26 years living in Massachusetts | 20 | Yes |
| Koskan 2018 [73] | "This study explores HIV-positive gay and bisexual men's (GBM) understanding of human papillomavirus (HPV) and the HPV vaccine." | United States | Community | Qualitative, Interviews | None Indicated | HIV-positive, self-identify as gay or bisexual men [transgender populations excluded] age 18 years and older, English or Spanish fluency, reside in Miami-Dade County). | 15 | No |
| Lee 2007 [74] | "To assess the knowledge and beliefs on cervical cancer and HPV infection and to evaluate the acceptability of HPV vaccination among Chinese women" | China | Not Specified/ Uncertain | Qualitative, Focus Groups | None Indicated | Healthy ethnic Hong Kong Chinese women aged 18 years and older | 49 | No |
| Lim 2019 [75] | "To explore the facilitators and barriers of HPV vaccination in young females aged 18–26 years in Singapore, and to describe their recommended strategies to improve the uptake of HPV vaccination" | Singapore | School | Qualitative, focus groups and interviews | Deductive Approach | Female students, aged 18–26 years old studying at the National University of Singapore (NUS) | 40 | Yes |
| Mancuso 2010 [76] | "learn more about how young women make decisions about HPV vaccination" | Canada | Not Specified/ Uncertain | Qualitative, Interviews | None Indicated | Five young women between the ages of 18 and 26 | 5 | Yes |
| Martin 2011 [77] | "The aim of this small-scale study is to explore the impact of the introduction of HPV vaccine on attitudes towards HPV, cervical cancer and sexual risk-taking amongst university students aged 20 to 24 years" | United Kingdom | Not Specified/ Uncertain | Qualitative, Focus Groups | None Indicated | Participants were male and female students, aged 20–24 and students at the University of Leeds | 34 | Yes |
| McClelland 2006 [78] | "This paper explores knowledge of and attitudes toward sexually transmissible infections, human papillomavirus (HPV) vaccination and vaccine acceptability among young people in Australia." | Australia | Community | Qualitative, Interviews | None Indicated | The inclusion criterion in relation to participants? Age was determined according to the target population of the current vaccination trials (18?23 years) because men and women often start their sexual lives at this age | 14 | Yes |

*(Continued)*

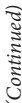

| Author & Year | Study Objective | Country | Study Setting | Design & Data Collection | Theoretical Framework | Eligible Participants | Sample Size | Purposive Sampling |
|---|---|---|---|---|---|---|---|---|
| McComb 2018 [79] | "The purpose of this study was to explore knowledge, attitudes and barriers regarding the HPV vaccine in this population." | Canada | Not Specified/ Uncertain | Qualitative, Interviews | Health Belief Model | Eligibility criteria included women who were born in a country other than Canada, without a prior history of vaccination against HPV and who were proficient in English | 11 | Yes |
| Mehta 2013 [80] | "The purpose of this study was to determine predictors of HPV vaccine acceptability among college-aged men" | United States | School | Qualitative, Focus Groups | Health Belief Model | Eligibility requirements for enrollment were: English speaking males, 18 years or older, attending the university of interest. The upper age limit for participation was 25 years | 50 | Yes |
| Miller-Day 2023 [81] | "The purpose of the current project is to first identify elements of young men's vaccine decision narratives (phase one) and then translate these into prevention messages for a "Men's Stories" (MS) narrative-based HPV video intervention (phase two)." | United States | Not Specified/ Uncertain | Qualitative, Interviews | Narrative Engagement Theory | Self-identify as male, age 18–26, English speaking, and self-identify as heterosexual. | 15 | Yes |
| Mills 2013 [82] | "To address women's reasons for declining the HPV vaccine and, among women who initiated the vaccine series, barriers to completion of the 3-dose regimen" | United States | Not Specified/ Uncertain | Qualitative, Interviews | Integrated Behavioral Model (IBM) | Female clinic patients aged 18–26, who had either declined the HPV vaccine or had started the vaccine series but failed to complete doses 2, 3, or both in the required time | 17 | Yes |
| Morales-Campos 2021 [83] | To evaluate "(1) What are the gendered beliefs and attitudes around cervical cancer, HPV and HPV vaccination among Mexican and Mexican American men and women living along the Texas-Mexico Border? And (2) What are similarities/differences between mothers and fathers beliefs who have HPV vaccine-eligible daughters?" | United States | Not Specified/ Uncertain | Qualitative, Focus Groups | None Indicated | Being a Hispanic man or woman, aged 18 or older, a parent of a girl aged 11–17 (only for mother and father groups), and residing in Hidalgo or Cameron counties. | 71 | No |
| Mortensen 2010 [84] | "Investigate the reasons for young women's acceptance or rejection of the quadrivalent HPV vaccine after its general availability in Denmark" | Denmark | Not Specified/ Uncertain | Qualitative, focus groups and interviews | None Indicated | Women aged between 16–26 years who had heard of HPV vaccination | 435 | No |
| Nadarzynski 2017 [85] | "explore the perceptions of HPV and attitudes towards HPV vaccination to inform the development of future interventions on HPV vaccination for MSM in the United Kingdom." | United Kingdom | Not Specified/ Uncertain | Qualitative, focus groups and interviews | None Indicated | "English-speaking MSM, between 16 and 40 years old. All self-identified men, who were sexually attracted to or had already had sex with other men, were eligible for inclusion in the study" | 32 | No |

*(Continued)*

Table 1. (Continued)

| Author & Year | Study Objective | Country | Study Setting | Design & Data Collection | Theoretical Framework | Eligible Participants | Sample Size | Purposive Sampling |
|---|---|---|---|---|---|---|---|---|
| Nkwonta 2020 [86] | "explore 1) international university students knowledge of HPV and its associated sequelae; and 2) international university students attitudes toward and uptake of HPV preventive practices." | United States | School | Mixed Methods, interview, focus group, and questionnaire | Health Belief Model | 1) were 18 years or older; 2) self-reported as an international university student; and 3) had spent at least a semester in the US. | 81 | No |
| Pčolkina 2019 [87] | "We explored knowledge, behaviors, and attitudes toward CC prevention strategies in Latvian women" | Latvia | Clinical | Mixed Methods, interview and questionnaire | None Indicated | Latvian women in Riga age 20 and older | 158 | No |
| Petrova 2015 [88] | "explore young women's experiences with risk information about HPV and the HPV vaccine. We explored how women evaluated the quantity and quality of HPV-related information and information sources they consulted, and how this information affected their decisions about the HPV vaccine" | Multiple Countries (Bulgaria, Scotland, Serbia, Spain) | School | Qualitative, Focus Groups | None Indicated | Women between 18 and 26 years old | 54 | Yes |
| Pierre Joseph 2014 [89] | "To examine the attitudes toward human papillomavirus (HPV) vaccination among young men from African American, Haitian, Caucasian, and Latino backgrounds" | United States | Community | Qualitative, Interviews | Health Belief Model | 89 men (31 black, 26 Haitian, 18 Latinos and 14 Caucasian), average age 19 | 89 | Yes |
| Pierre-Victor 2017 [90] | "This study investigated the role of healthcare providers' Recommendation style in Haitian parents' and female patients' HPV vaccine decision-making" | United States | School | Qualitative, Interviews | None Indicated | Self-identifying as Haitian, female, and being between 17 and 26 years of age. | 30 | No |
| Power 2009 [91] | "explore how lesbian and bisexual women perceive their level of risk for HPV, their willingness to accept the HPV vaccine, the reasons why they do or do not feel at risk and how they manage their sexual health in relation to their identity as a non-heterosexual woman" | Australia | School | Mixed Methods, interview and questionnaire | Symbolic Interactionist Theory | Lesbian and bisexual women and a comparison group of heterosexual women | 352 | No |
| Pratt 2019 [92] | "This study sought to understand the views of Somali young adults regarding HPV immunization." | Somalia | Clinical | Qualitative, Focus Groups | None Indicated | Somali and between 18 and 26 years of age | 34 | Yes |

(Continued)

Table 1. (Continued)

| Author & Year | Study Objective | Country | Study Setting | Design & Data Collection | Theoretical Framework | Eligible Participants | Sample Size | Purposive Sampling |
|---|---|---|---|---|---|---|---|---|
| Prokopovich 2023 [93] | "explore how one culturally and linguistically diverse (CALD) community engages with school and HPV vaccination" | Australia | Not Specified/ Uncertain | Qualitative, Focus Groups | None Indicated | "(I) parents or grandparents of a child/grandchild aged 7–24 years old (those who have consented or will be approached to consent for school vaccination); and (ii) young adults aged 18?24 years old who had been exposed to their local school vaccination programme." | 31 | No |
| Rahim 2023 [94] | "The objective of this study was to assess vaccination rates in both populations at our own institution as well as understand AYA and caregiver's beliefs regarding HPV vaccination and vaccine hesitancy, which may contribute to the lower HPV vaccination rates in these high-risk patient populations." | United States | Clinical | Mixed Methods, Retrospective chart review and interview | None Indicated | Caregivers of patients 9–21 years of age and patients who were 18–21 years old who presented to either the comprehensive SCD clinic or the oncology survivorship clinic from March 2021 to July 2021 | 69 | No |
| Reiter 2014 [95] | "The current study collected data from four key stakeholder groups from Appalachian communities to examine their acceptability of HPV vaccine for males and potential barriers to vaccinating males against HPV in their communities." | United States | Not Specified/ Uncertain | Qualitative, focus groups and interviews | None Indicated | (A) parents with adolescent sons ages 9–17 years, (b) young adult men ages 18–26 years (c) health care providers, and (d) community leaders. | 102 | No |
| Ross 2010 [96] | "We undertook a qualitative study among first year university students as a first step towards developing a model that explains pathways to vaccine acceptance or rejection amongst older teenagers." | United Kingdom | School | Qualitative, Focus Groups | Multiple Theories (Health Belief model, Social Cognitive Model) | "female first year undergraduates who resided permanently in the UK and were therefore eligible for HPV vaccination before they came to University. Undergraduates studying nursing and medicine were not eligible for the study." | 0 | Yes |
| Salad 2015 [97] | "aims to explore the perceptions of Somali women living in the Netherlands regarding measures to prevent cervical cancer" | Netherlands | Not Specified/ Uncertain | Qualitative, focus groups and interviews | Multiple Theories (Health Belief model, intersectionality framework) | "being female and of Somali origin, living in the Netherlands, aged between 18 and 65, and having a migration date from the first or second wave of migration" | 20 | No |
| Schmidt-Grimminger 2013 [98] | "We assessed HPV knowledge, attitudes, and beliefs towards HPV and HPV vaccination during a community-based participatory research project among tribal youth, young adults, parents, and health professionals" | United States | Not Specified/ Uncertain | Mixed Methods, focus group and questionnaire | CBPR Framework | Young adult women ages 19–26 (n=22). | 73 | No |

*(Continued)*

| Author & Year | Study Objective | Country | Study Setting | Design & Data Collection | Theoretical Framework | Eligible Participants | Sample Size | Purposive Sampling |
|---|---|---|---|---|---|---|---|---|
| Schwendener 2022 [99] | "We aimed to provide a detailed characterisation of human papillomavirus (HPV) vaccine awareness, knowledge and information sources in the HPV vaccine decision-making process of youth, both male and female, in Switzerland." | Switzerland | Not Specified/ Uncertain | Mixed Methods, interview and questionnaire | None Indicated | Participants were 15–26 years of age, male and female | 997 | Unclear/Not Specified |
| Shetty 2021 [100] | "The purpose of this exploratory study is to evaluate undergraduate student understanding towards barriers and facilitators of HPV vaccination and identify potential health education opportunities for HPV vaccination." | India | School | Qualitative, Focus Groups | None Indicated | "students aged 18–26 years enrolled in medical, dental, nursing, and MLT degree courses at KSHEMA who previously participated in a previously published quantitative study" | 20 | Yes |
| Siu 2013 [101] | "This paper investigates, using a qualitative approach, barriers to receiving Human Papillomavirus (HPV) vaccine among female undergraduate students in a Hong Kong university." | China | School | Qualitative, Interviews | None Indicated | (1) female undergraduate students, excluding medical and health sciences students, (2) no experience of sexual intercourse, (3) have not yet received an HPV vaccination and (4) Hong Kong Chinese by ethnicity | 35 | Yes |
| Stephens 2014 [102] | "assess current and preferred social networks that influence human papillomavirus(HPV) vaccine decision-making in a sample of Hispanic college women." | United States | School | Qualitative, Interviews | Social Network Theory | Criteria included self-identifying as Hispanic, female, and between 18 and 24 years of age | 41 | Yes |
| Stephens 2016 [103] | "To identify factors influencing human papillomavirus (HPV) vaccination up taking decision making among vaccinated and non-vaccinated Hispanic college women." | United States | School | Qualitative, Interviews | None Indicated | Hispanic undergraduate women aged 18–24 | 49 | Yes |
| Sullivan-Blum 2022 [104] | "This study aimed to determine the attitudes, beliefs and barriers of primary care patients on prep towards the HPV vaccine." | United States | Clinical | Qualitative, Interviews | None Indicated | Status as a prep patient in the two family medicine clinics | 16 | No |
| Teitelman 2018 [105] | "the purpose of this article is to identify salient beliefs about HPV vaccine completion among young adult women who live in economically disadvantaged urban communities and describe the integration of those beliefs into the development of an mhealth application using theory-based research and user-centered design to promote vaccine completion." | United States | Community | Mixed Methods, Survey/ Questionnaire | Integrated Behavioral Model (IBM) | "women between 18 and 26 years of age who received the first dose of the HPV vaccine at the current clinic visit, had ever had vaginal sex, and owned a smartphone." | 35 | Yes |

*(Continued)*

**Table 1.** (Continued)

| Author & Year | Study Objective | Country | Study Setting | Design & Data Collection | Theoretical Framework | Eligible Participants | Sample Size | Purposive Sampling |
|---|---|---|---|---|---|---|---|---|
| Thompson 2017 [106] | "The purpose of this study was to understand how relationship status and vaccination status impact risk perceptions and perceived need for the HPV vaccine among young adult women." | United States | School | Mixed Methods, interview and questionnaire | None Indicated | "Women were eligible for the study if they met the following criteria: 1) university student, 2) between 18 and 26 years of age, 3) has not received any doses of the HPV vaccine or has received the first dose of the HPV vaccination series in the last 6 months, 4) speaks English, and 5) provides informed consent." | 50 | Yes |
| Thompson 2018 [107] | "The purpose of this study was to elicit the information needs, motivations, and behavioral skills related to HPV vaccine decision-making among young adult women by comparing these factors between unvaccinated and recently vaccinated women." | United States | School | Mixed Methods, interview and questionnaire | Information, Motivation, and Behavioral Skills Model (IMB) | (1) 18–26 years of age, (2) college student, (3) had not received the HPV vaccine or received the first dose of the HPV vaccine series in the last 6 months, and (4) English speaking | 50 | Yes |
| Tu 2013 [108] | "To enhance understanding of young women's experiences of human papillomavirus-related cervical cancer prevention in Taiwan" | Taiwan | School | Qualitative, Interviews | None Indicated | "The college women participants met the criteria below: (1) above 20 years old with sexual experience and never been married and/or pregnant, (2) heard about HPV and (3) were willing to share their HPV-related cervical cancer prevention experiences" | 24 | Yes |
| Waters 2022 [109] | "We sought to describe participants' experiences with the HPV vaccine" | United States | Clinical | Mixed Methods, interview and questionnaire | None Indicated | "Eligible participants were either an HPV vaccine eligible cancer survivor (18–26 years) or a caregiver of a younger HPV vaccine eligible cancer survivor (9-17 years). Eligible survivors had completed treatment and received care during 2013-2018. Eligible caregivers were at least 18 years of age and the caregiver of a survivor under 18 years of age who had completed treatment at PCH between 2013 and 2018" | 20 | No |
| Wheldon 2017 [110] | "The purpose of this study was to (1) describe salient beliefs related to HPV vaccination among young MSM; (2) determine factors that underlie these beliefs; and (3) describe a model for HPV vaccine decision-making." | United States | Community | Qualitative, Interviews | Integrative Model of Behavioral Prediction (IM) | MSM between the ages of 18 and 26 years who may or may not identify as gay or bisexual | 22 | Yes |

*(Continued)*

**Table 1.** (Continued)

| Author & Year | Study Objective | Country | Study Setting | Design & Data Collection | Theoretical Framework | Eligible Participants | Sample Size | Purposive Sampling |
|---|---|---|---|---|---|---|---|---|
| Wong 2008 [111] | "To investigate the acceptability of the HPV vaccine among a multi-ethnic sample of young women in Malaysia" | Malaysia | Not Specified/ Uncertain | Qualitative, Focus Groups | None Indicated | Young unmarried Malaysian women aged between 13 and 27 years. | 40 | No |
| Wong 2009 [112] | "This qualitative study used focus group discussions (FGDs) to evaluate information needed in order to make informed human papillomavirus (HPV) vaccination decision, opinion on the most acceptable public education messages, and channel of delivery in a multiethnic, multicultural and multireligion country." | Malaysia | Community | Qualitative, Focus Groups | Grounded Theory | Mothers of eligible vaccinees, young women eligible for the vaccine aged between 18 and 26 years old, and men (single men and fathers of eligible vaccinees). | 114 | No |
| Wyndham-West 2016 [113] | "In this article I explore the limitation of the public health account of decision-making, grounded in individual cost/benefit rationality and explore the ways in which for these students vaccination decisions were shaped by social context and identity issues" | Canada | School | Qualitative, Interviews | None Indicated | Ontario-based women university students who were up to 26 years of age. | 24 | Yes |
| Yarmoham-madi 2022 [114] | "this study aimed to determine strategies for improving HPV vaccination among young adults" | Iran | Community | Qualitative, Interviews | None Indicated | (1) Tehran native young adult participant, (2) 18–26 years old, and (3) having information about HPV (both young adult participants and health professionals) | 30 | No |
| Young 2018 [115] | "The objective of this study is to determine the knowledge, understanding and concerns that young women have about HPV when attending colposcopy and whether their information needs are met." | United Kingdom | Clinical | Qualitative, Interviews | None indicated | "all English-speaking women born after 1/9/1990 with abnormal cervical cytology and who had been referred to colposcopy at a regional colposcopy clinic were eligible to take part" | 15 | Yes |

**Table 2. Study Characteristics Comparison Between Eligible and Included Studies.**

| | Eligible Studies | Included Studies |
|---|---|---|
| **# Studies (articles)** | 68 studies (71 articles) | 42 studies (45 articles) |
| **# Participants** | 4788 | 1790 |
| **Study Design** | Qualitative: 56 | Qualitative: 37 |
| | Mixed Methods: 12 | Mixed Methods: 5 |
| **Data Collection** | Between 2006 and 2021 | Between 2007 and 2021 |
| **Study Setting** | Clinical: 7 | Clinical: 3 |
| | Community: 12 | Community: 8 |
| | School: 26 | School: 21 |
| | Not specified: 23 | Not specified: 10 |
| **Continent & Country** | 5 continents & 19 countries | 5 continents & 14 countries |
| | Africa (1): Somalia (1) | Africa (1): Somalia (1) |
| | Asia (12): China (4), India (1), Iran (1), Malaysia (3), Singapore (2), Taiwan (1) | Asia (8): China (3), India (1), Malaysia (1), Singapore (2), Taiwan (1) |
| | Europe (10): Denmark (1), Latvia (1), Netherlands (1), Switzerland (1), United Kingdom (5); Multiple countries (1) | Europe (4): United Kingdom (3); Multiple countries (1) |
| | North America (41): Canada (3), United States (38) | North America (28): Canada (3), United States (25) |
| | Oceania(4): Australia (4) | Oceania(1): Australia (1) |
| **Economy Classification** | High income: 13 countries | High income: 9 countries |
| | Middle-income: 5 countries | Middle-income: 4 countries |
| | Low-income: 1 country | Low-income: 1 country |
| **Study Participants: groups** | Women: 55 | Women: 34 |
| | Men: 30 | Men: 17 |
| | SGM: 7 | SGM: 3 |
| | College students: 16 | College students: 14 |
| | Parents: 4 | |
| | Healthcare professionals: 4 | |
| | Caregivers: 2 | |
| | Community leaders: 1 | |
| **Study Participants: Race and Ethnicity (as reported from the articles; not mutually exclusive)** | White/Caucasian: 21 | White/Caucasian: 15 |
| | Non-Hispanic White: 4 | Non-Hispanic White: 1 |
| | Non-Hispanic/Latinx: 1 | Non-Hispanic/Latinx: 1 |
| | Hispanic White: 2 | Hispanic White: 1 |
| | Hispanic/Latinx: 20 | Hispanic/Latinx: 15 |
| | Mexican American: 1 | Mexican American: 1 |
| | Black/African American: 20 | Black/African American: 16 |
| | Asian: 12 | Asian: 8 |
| | Asian American: 1 | Asian American: 1 |
| | Native American: 3 | Native American: 2 |
| | Multiracial/Multiethnic: 5 | Multiracial/Multiethnic: 3 |
| | Other/Unknown: 12 | Other/Unknown: 8 |
| **Article Publication** | Between 2006 and 2023 | Between 2006 and 2023 |

**Theoretical frameworks & Conceptual models.** Nineteen (19) theoretical frameworks and conceptual models were utilized across the eligible articles. The Health Belief Model was the most common theory cited (7 studies). Thirty-eight (38) articles did not cite use of a theory or model.

**Study setting and country.** Twenty-six (26) studies cited the study was conducted in a school setting, 12 in a community setting and 7 in a clinical setting; 23 studies did not specify a study setting. Across the eligible articles, 19 countries spanning 5 continents were represented; according to the World Bank classification of economies [116], 13 were high income countries (Australia, Canada, Bulgaria, Denmark, Latvia, Netherlands, Scotland, Singapore, Spain, Switzerland, Taiwan, United Kingdom, United States); 5 were middle income countries (China, India, Iran, Malaysia, Serbia); and 1 was a low-income country (Somalia). Most articles presented studies conducted in North America (41 studies), majority of which were in the United States (38 studies); 12 studies were conducted in Asia, 10 in Europe, 4 in Oceania, and only one 1 study was conducted in Africa. One study spanned multiple countries in Europe (Bulgaria, Scotland, Serbia, and Spain).

**Study participants.** There were a total of 4788 participants across the eligible studies. Fifty-five (55) studies sampled women/females, 30 sampled men/males, and 7 sampled sexual and gender minority (SGM) individuals. College students were represented in 16 studies, parents were sampled in 4 studies, healthcare professionals were included in 4 studies, caregivers sampled in 2 studies, and community leaders were included in 1 study.

Race and ethnicity categories were not uniformly reported across articles. We present them below as they were reported in the manuscripts, acknowledging that some labels may seem duplicitous/refer to the same classification of race/ethnicity: 21 studies included White/Caucasian individuals; 4 studies included Non-Hispanic White individuals; 1 study included Non-Hispanic/Latinx individuals; 2 studies included Hispanic White individuals; 20 studies included Hispanic/Latinx individuals; 1 study included Mexican American individuals; 20 studies included Black/African American individuals; 12 studies included Asian individuals; 1 study included Asian American individuals; 3 studies included Native American/American Indian individuals; 5 studies included multiracial/multiethnic individuals; and 12 studies cited individuals with other/unknown race and ethnicity.

**Study publication.** Study manuscripts were published between 2006 and 2023

Table 2 provides a summarized comparison of the study characteristics between the 68 studies that met review eligibility and the 42 studies that were included in the review after purposive sampling.

## Methodological quality of included studies

Across the 42 studies that were purposively sampled and included in the review, the majority (35 studies represented by 37 articles) had an overall assessment of minor to moderate quality (Table 3) The most common issue across the studies was poor or no reporting of researcher reflexivity (36 studies). It is worth noting that the requirement for reflexivity in reporting guidelines for qualitive research may have been implemented after some of the studies were published. Additionally, three studies did not provide details on data analysis procedures and failed to take ethical issues into consideration (no mention of informed consent or IRB approval) [46,55,76]. Of the 42 included studies, only two studies had severe concerns on methodological quality [46,55].

## Confidence in review findings

Our QES yielded 29 review findings. We graded 11 findings as high confidence, 17 findings as moderate confidence and 1 finding as low confidence (Table 4). The GRADE CERQual evidence profile detailing our evaluation of each component for each finding and associated explanations are in S3 Appendix.

## Synthesis of findings

This section details the review findings from the QES. The review findings are categorized into 10 thematic categories that emerged from the thematic data analysis process to reflect the barriers and facilitators to HPV vaccine uptake and

**Table 3. Methodological limitations of included studies.**

| Study ID | Was there a clear statement of the aims of the research? | Are the setting(s) and context described adequately? | Is the sampling strategy described, and is this appropriate? | Is the sampling strategy described, and is this appropriate? | Is the data analysis described, and is this appropriate? | Is there evidence of reflexivity? | Have ethical issues been taken into consideration? | Are the claims made/findings supported by sufficient evidence? | Overall assessment |
|---|---|---|---|---|---|---|---|---|---|
| Allen 2009 | Yes | Yes | Can't Tell | Yes | Yes | No | Yes | Yes | Moderate |
| Al-Naggar 2010 | Yes | Yes | Yes | Yes | No | No | No | Yes | Severe |
| Apaydin 2018 | Yes | Yes | Yes | Yes | Yes | No | Yes | Yes | Minor to moderate |
| Basnyat 2018 | Yes | Yes | Yes | Yes | Yes | No | Yes | Yes | Minor to moderate |
| Chan 2011 | Yes | Yes | Yes | Yes | Yes | Can't Tell | Yes | Yes | Minor to moderate |
| Chen 2021 | Yes | Yes | Yes | Yes | Yes | No | Yes | Yes | Minor to moderate |
| Clevenger 2012 | Yes | Yes | Yes | Yes | Yes | No | Yes | Yes | Minor to moderate |
| Cohen 2013 | Yes | Yes | Yes | Yes | Yes | Can't Tell | Yes | Yes | Minor to moderate |
| Dai 2020 | No | Yes | No | No | No | Yes | No | Yes | Severe |
| Fontenot 2016 | Yes | Yes | Yes | Yes | Yes | No | Yes | Yes | Minor to moderate |
| Garcia 2023 | Yes | Yes | Yes | Yes | Yes | No | Yes | Yes | Minor to moderate |
| Gerend 2019 | Yes | Yes | Yes | Yes | Yes | No | Yes | Yes | Minor to moderate |
| Gray Brunton 2014 | Yes | Yes | Yes | Yes | Yes | No | Yes | Yes | Minor to moderate |
| Head 2012 | Yes | Yes | Yes | Yes | Yes | No | Yes | Yes | Minor to moderate |
| Hirth 2018 | Yes | Yes | Yes | Yes | Yes | No | Yes | Yes | Minor to moderate |
| Hodge 2011 | Yes | Yes | Yes | Yes | Yes | Yes | Yes | Yes | Minor |
| Hodge 2014 | Yes | Yes | Yes | Yes | Yes | No | Yes | Yes | Minor to moderate |
| Hopfer 2011 | Yes | Yes | Yes | Yes | Yes | Yes | Yes | Yes | Minor |
| Jaiswal 2020 | Yes | Yes | Yes | Yes | Yes | No | Yes | Yes | Minor to moderate |
| Jin 2023 | Yes | Yes | Yes | Yes | Yes | No | Yes | Yes | Minor to moderate |
| Joseph 2014 | Yes | Yes | Yes | Yes | Yes | No | Yes | Yes | Minor to moderate |
| Kim 2017 | Yes | Yes | Yes | Yes | Yes | No | Yes | Yes | Minor to moderate |
| Lim 2019 | Yes | Yes | Yes | Yes | Yes | Yes | Yes | Yes | Minor |
| Mancuso 2010 | Yes | Yes | Yes | Yes | No | No | No | Yes | Moderate to severe |
| Martin 2011 | Yes | Yes | Yes | Yes | Yes | No | Yes | Yes | Minor to moderate |
| McClelland 2006 | Yes | Yes | Yes | Yes | Yes | No | Yes | Yes | Minor to moderate |
| McComb 2018 | Yes | Yes | Yes | Yes | Yes | No | Yes | Yes | Minor to moderate |
| Mehta 2013 | Yes | Yes | Yes | Yes | Yes | No | Yes | Yes | Minor to moderate |
| Miller-Day 2023 | Yes | Yes | Yes | Yes | Yes | No | Yes | Yes | Minor to moderate |
| Mills 2013 | Yes | Yes | Yes | Yes | Yes | No | Yes | Yes | Minor to moderate |
| Petrova 2015 | Yes | Yes | Yes | Yes | Yes | No | Yes | Yes | Minor to moderate |
| Pierre Joseph 2014 | Yes | Yes | Yes | Yes | Yes | No | Yes | Yes | Minor to moderate |
| Pratt 2019 | Yes | Yes | Yes | Yes | Yes | No | Yes | Yes | Minor to moderate |
| Ross 2010 | Yes | Yes | Yes | Yes | Yes | No | Yes | Yes | Minor to moderate |
| Shetty 2021 | Yes | Yes | Yes | Yes | Yes | No | Yes | Yes | Minor to moderate |
| Siu 2013 | Yes | Yes | Yes | Yes | Yes | No | Yes | Yes | Minor to moderate |

*(Continued)*

**Table 3.** (Continued)

| Study ID | Was there a clear statement of the aims of the research? | Are the setting(s) and context described adequately? | Is the sampling strategy described, and is this appropriate? | Is the sampling strategy described, and is this appropriate? | Is the data analysis described, and is this appropriate? | Is there evidence of reflexivity? | Have ethical issues been taken into consideration? | Are the claims made/findings supported by sufficient evidence? | Overall assessment |
|---|---|---|---|---|---|---|---|---|---|
| Stephens 2014 | Yes | Yes | Yes | Yes | Yes | No | Yes | Yes | Minor to moderate |
| Stephens 2016 | Yes | Yes | Yes | Yes | Yes | Yes | Yes | Yes | Minor |
| Teitelman 2018 | Yes | Yes | Yes | Yes | Yes | No | Yes | Yes | Minor to moderate |
| Thompson 2017 | Yes | Yes | Yes | Yes | Yes | No | Yes | Yes | Minor to moderate |
| Thompson 2018 | Yes | Yes | Yes | Yes | Yes | No | Yes | Yes | Minor to moderate |
| Tu 2013 | Yes | Yes | Yes | Yes | Yes | No | Yes | Yes | Minor to moderate |
| Wheldon 2017 | Yes | Yes | Yes | Yes | Yes | No | Yes | Yes | Minor to moderate |
| Wyndham-West 2016 | Yes | Yes | Yes | Yes | Yes | No | Yes | Yes | Minor to moderate |
| Young 2018 | Yes | Yes | Yes | Yes | Yes | No | Yes | Yes | Minor to moderate |

**Table 4. Summary of qualitative finding.**

| Summarized review finding | GRADE-CERQual Assessment of confidence | Explanation of GRADE-CERQual Assessment | References |
|---|---|---|---|
| **THEME 1: Individual Factors** | | | |
| [Finding 1] Missed opportunity: Some young adults believed they missed the window for vaccination because they were too old to benefit from it | Moderate confidence | 5 studies reflecting minor concerns regarding methodological limitations, and minor concerns regarding adequacy. | Kim 2017; Jaiswal et al. 2020; Tu & Wang 2013; Wheldon et al. 2017; Hodge et al. 2011; |
| [Finding 2] Risk perception: Some young adults believed they were not at risk for HPV-related illnesses or that they still had time to decide on vaccination. | Moderate confidence | 7 studies reflecting minor concerns regarding methodological limitations, minor concerns regarding coherence, and minor concerns regarding adequacy. | Hodge 2014; Hopfer & Clippard 2011; Allen et al. 2009; Lim & Lim 2019; Miller-Day et al. 2023; Wheldon et al. 2017; Siu 2013; |
| [Finding 3] Proactive measure: Young adults viewed vaccination as a proactive measure to safeguard their health in the present and prevent future regret if they didn't get vaccinated. | High confidence | 13 studies reflecting minor concerns regarding methodological limitations. | Hopfer & Clippard 2011; Apaydin et al. 2018; Chen et al. 2021; Gerend et al. 2019; Hirth et al. 2018; Joseph et al. 2014; Pierre Joseph et al. 2014; Shetty et al. 2021; Thompson et al. 2017; Wheldon et al. 2017; Wyndham-West 2016; McClelland & Liamputtong 2006; Stephens et al. 2016; |
| [Finding 4] Low-stakes preventive measure: Several individuals mention the simplicity of the decision to get vaccinated, seeing it as a low-risk preventive measure with potentially significant benefits | Low confidence | 5 studies reflecting minor concerns regarding methodological limitations, and moderate concerns regarding adequacy. | Kim 2017; Hirth et al. 2018; Lim & Lim 2019; Pierre Joseph et al. 2014; Cohen & Head 2013; McClelland & Liamputtong 2006; |
| **THEME 2: Alternatives to vaccination** | | | |
| [Finding 5] Reliance on pap smears: Young adult women believed that regular Pap testing negated the need for HPV vaccination. | Moderate confidence | 4 studies reflecting moderate concerns regarding methodological limitations, and minor concerns regarding adequacy. | Mancuso & Polzer 2010; Petrova et al. 2015; Tu & Wang 2013; Hodge et al. 2011; |
| [Finding 6] Condom use and other precautions: Young adults expressed precautions such as condom use, abstinence, and decisions on sexual partners were sufficient to protect against HPV, rendering the vaccine unnecessary. | Moderate confidence | 6 studies reflecting minor concerns regarding methodological limitations, and minor concerns regarding adequacy. | Hopfer & Clippard 2011; Miller-Day et al. 2023; Pierre Joseph et al. 2014; Thompson et al. 2017; Tu & Wang 2013; Cohen & Head 2013; |
| **THEME 3: Knowledge and information** | | | |
| [Finding 7] Many young adults expressed having insufficient information on HPV or the vaccine before deciding to get vaccinated. There was an expressed desire more information about HPV and the HPV vaccine. | High confidence | 13 studies reflecting minor concerns regarding methodological limitations. | Hodge 2014; Kim 2017; Fontenot et al. 2016; Garcia et al. 2023; Gray Brunton et al. 2014; Hirth et al. 2018; Joseph et al. 2014; Miller-Day et al. 2023; Pierre Joseph et al. 2014; Cohen & Head 2013; Head & Cohen 2012; Hodge et al. 2011; McClelland & Liamputtong 2006; |
| **THEME 4: Sex and romantic relationships** | | | |
| [Finding 8] Sexual activity and behavior: Some individuals believe that vaccination against HPV is only necessary if one plans to engage in sexual activity or has many sexual partners. | High confidence | 12 studies reflecting minor concerns regarding methodological limitations. | Hopfer & Clippard 2011; Garcia et al. 2023; Joseph et al. 2014; Lim & Lim 2019; Miller-Day et al. 2023; Petrova et al. 2015; Thompson et al. 2017; Wheldon et al. 2017; Cohen & Head 2013; Head & Cohen 2012; Siu 2013; Stephens et al. 2016; |
| [Finding 9] Relationship status and sexual history influenced risk perception among young adults. | High confidence | 7 studies reflecting minor concerns regarding methodological limitations. | Hopfer & Clippard 2011; Miller-Day et al. 2023; Pierre Joseph et al. 2014; Thompson et al. 2017; Thompson et al. 2018; McClelland & Liamputtong 2006; Mehta et al. 2013; |

*(Continued)*

| Summarized review finding | GRADE-CERQual Assessment of confidence | Explanation of GRADE-CERQual Assessment | References |
|---|---|---|---|
| [Finding 10] Parental beliefs about sex and the HPV vaccine influenced young adults' vaccine attitudes and decisions. | Moderate confidence | 6 studies reflecting minor concerns regarding methodological limitations, minor concerns regarding coherence, minor concerns regarding adequacy. | Hopfer & Clippard 2011; Fontenot et al. 2016; McComb et al. 2018; Shetty et al. 2021; Hodge et al. 2011; Siu 2013; |
| **THEME 5: Parents and peers** | | | |
| [Finding 11] Mothers more than fathers played an active role in HPV vaccination process among young adults. | High confidence | 12 studies reflecting minor concerns regarding methodological limitations. | Hopfer & Clippard 2011; Kim 2017; Garcia et al. 2023; Gerend et al. 2019; Hirth et al. 2018; Lim & Lim 2019; Miller-Day et al. 2023; Basnyat & Lim 2018; Cohen & Head 2013; McClelland & Liamputtong 2006; Ross et al. 2010; Stephens & Thomas 2014; |
| [Finding 12] Decision-making power: There was a mix of deferred or shared the decision to get vaccinated between young adults and parents. | Moderate confidence | 5 studies reflecting minor concerns regarding methodological limitations, minor concerns regarding coherence, and minor concerns regarding adequacy. | Mills et al. 2013; Hirth et al. 2018; Miller-Day et al. 2023; Cohen & Head 2013; Stephens et al. 2016; |
| [Finding 13] Peer influence: Friends and acquaintances played a significant role in shaping vaccination decisions. | Moderate confidence | Minor concerns regarding methodological limitations, No/Very minor concerns regarding coherence, Minor concerns regarding adequacy, and No/Very minor concerns regarding relevance | Hopfer & Clippard 2011; Thompson et al. 2018; Wheldon et al. 2017; Basnyat & Lim 2018; Cohen & Head 2013; Head & Cohen 2012; Ross et al. 2010; Siu 2013; |
| [Finding 14] Health history of family and friends was a motivating factor in the decision to get vaccinated. | Moderate confidence | 5 studies reflecting minor concerns regarding methodological limitations and minor concerns regarding adequacy. | Chen et al. 2021; Pierre Joseph et al. 2014; Wyndham-West 2016; Head & Cohen 2012; Stephens et al. 2016; |
| **THEME 6: Physicians** | | | |
| [Finding 15] Doctor recommendation: Physician recommendations (positive or negative) were influential in vaccination uptake. | High confidence | 18 studies reflecting minor concerns regarding methodological limitations. | Hopfer & Clippard 2011; Apaydin et al. 2018; Gerend et al. 2019; Hirth et al. 2018; Jaiswal et al. 2020; Joseph et al. 2014; McComb et al. 2018; Miller-Day et al. 2023; Pierre Joseph et al. 2014; Thompson et al. 2017; Thompson et al. 2018; Wheldon et al. 2017; Wyndham-West 2016; Chan et al. 2011; Clevenger et al. 2012; Cohen & Head 2013; Ross et al. 2010; Stephens & Thomas 2014; |
| [Finding 16] Communication gap with doctors: Young adults desired better communication and more informed conversations with their doctors. | High confidence | 9 studies reflecting minor concerns regarding methodological limitations. | Mills et al. 2013; Apaydin et al. 2018; Fontenot et al. 2016; Garcia et al. 2023; Jaiswal et al. 2020; Petrova et al. 2015; Pierre Joseph et al. 2014; Wheldon et al. 2017; Siu 2013; |
| **THEME 7: Logistics** | | | |
| [Finding 17] Appointments: Scheduling and appointment challenges impacted young adults' ability to get vaccinated. | Moderate confidence | 10 studies reflecting minor concerns regarding methodological limitations and minor concerns regarding adequacy. | Mills et al. 2013; Apaydin et al. 2018; Fontenot et al. 2016; Hirth et al. 2018; Jaiswal et al. 2020; Pierre Joseph et al. 2014; Teitelman et al. 2018; Head & Cohen 2012; Stephens et al. 2016; Stephens & Thomas 2014; |
| [Finding 18] Multiple shots: Some young adults found the process of going back multiple times for the three doses burdensome, leading to forgetfulness or avoidance. | Moderate confidence | 8 studies reflecting minor concerns regarding methodological limitations and minor concerns regarding adequacy. | Hopfer & Clippard 2011; Apaydin et al. 2018; Jaiswal et al. 2020; Lim & Lim 2019; Miller-Day et al. 2023; Teitelman et al. 2018; Head & Cohen 2012; Ross et al. 2010; |

*(Continued)*

| Summarized review finding | GRADE-CERQual Assessment of confidence | Explanation of GRADE-CERQual Assessment | References |
|---|---|---|---|
| [Finding 19] Accessibility: Transportation and access challenges hindered HPV vaccination uptake. | Moderate confidence | 8 studies reflecting moderate concerns regarding methodological limitations and minor concerns regarding adequacy. | Dai 2020; Hopfer & Clippard 2011; Mills et al. 2013; Hirth et al. 2018; Lim & Lim 2019; Wheldon et al. 2017; Hodge et al. 2011; Ross et al. 2010; |
| [Finding 20] Reminders: Reminders were cited as helpful in ensuring adherence to the vaccination schedule. | Moderate confidence | 5 studies reflecting minor concerns regarding methodological limitations ad minor concerns regarding adequacy. | Hopfer & Clippard 2011; Apaydin et al. 2018; Hirth et al. 2018; Miller-Day et al. 2023; Ross et al. 2010; |
| [Finding 21] Mandates: Young adults had mixed views on mandates for HPV vaccination. | Moderate confidence | 7 studies reflecting minor concerns regarding methodological limitations and minor concerns regarding adequacy. | Hirth et al. 2018; Joseph et al. 2014; Lim & Lim 2019; Miller-Day et al. 2023; Pierre Joseph et al. 2014; Pratt et al. 2019; Shetty et al. 2021; |
| **THEME 8: The Vaccine** | | | |
| [Finding 22] Vaccine cost: Young adults expressed concerns about paying for the HPV vaccine, citing high out-of-pocket costs and need for government subsidies. | High confidence | 18 studies reflecting minor concerns regarding methodological limitations. | Hopfer & Clippard 2011; Chen et al. 2021; Gray Brunton et al. 2014; Hirth et al. 2018; Lim & Lim 2019; Miller-Day et al. 2023; Pierre Joseph et al. 2014; Shetty et al. 2021; Teitelman et al. 2018; Tu & Wang 2013; Wheldon et al. 2017; Chan et al. 2011; Clevenger et al. 2012; Cohen & Head 2013; Head & Cohen 2012; Hodge et al. 2011; Ross et al. 2010; Siu 2013; |
| [Finding 23] Side effects: Young adults expressed concerned about the side effects of the HPV vaccine. | High confidence | 12 studies reflecting minor concerns regarding methodological limitations. | Kim 2017; Mancuso & Polzer 2010; Garcia et al. 2023; Hirth et al. 2018; Jaiswal et al. 2020; Joseph et al. 2014; Petrova et al. 2015; Chan et al. 2011; Cohen & Head 2013; Hodge et al. 2011; Siu 2013; Stephens et al. 2016; |
| [Finding 24] Understanding of vaccine effectiveness: Young adults questioned the effectiveness of the HPV vaccine given the limited strains it covers and lack of 100% protection. | Moderate confidence | 9 studies reflecting minor concerns regarding methodological limitations, minor concerns regarding coherence, and minor concerns regarding adequacy. | Mancuso & Polzer 2010; Gray Brunton et al. 2014; Petrova et al. 2015; Thompson et al. 2017; Wheldon et al. 2017; Cohen & Head 2013; Hodge et al. 2011; Ross et al. 2010; Young et al. 2018; |
| [Finding 25] Development of the vaccine: Concerns were raised about the vaccine's novelty, hastened development, and lack of long-term studies. | High confidence | 11 studies reflecting minor concerns regarding methodological limitations. | Hodge 2014; Hopfer & Clippard 2011; Gray Brunton et al. 2014; Hirth et al. 2018; Lim & Lim 2019; Petrova et al. 2015; Tu & Wang 2013; Wyndham-West 2016; Chan et al. 2011; Cohen & Head 2013; Young et al. 2018; |
| [Finding 26] Pharmaceutical companies and business motive: Young adults expressed skepticism towards pharmaceutical companies and their motives, suggesting that public panic and greed may influence vaccination campaigns. | Moderate confidence | 5 studies reflecting minor concerns regarding methodological limitations and Minor concerns regarding adequacy. | Mancuso & Polzer 2010; Gray Brunton et al. 2014; Petrova et al. 2015; Chan et al. 2011; Siu 2013; |
| **Theme 9: Gendered perceptions and bias** | | | |
| [Finding 27] Gendered perceptions of HPV risk: Many participants exhibit misconceptions about HPV being primarily a women's disease. | Moderate confidence | 7 studies reflecting moderate concerns regarding methodological limitations and minor concerns regarding adequacy. | Hodge 2014; Kim 2017; Allen et al. 2009; Fontenot et al. 2016; Gerend et al. 2019; Jaiswal et al. 2020; Mehta et al. 2013; |
| [Finding 28] Gender biased promotion: The gendered marketing and promotion of HPV vaccines led to perceived unfairness in who gets access to vaccine and who is responsible for getting the vaccinated in the context of sexual health. | High confidence | 8 studies reflecting minor concerns regarding methodological limitations. | Hodge 2014; Apaydin et al. 2018; Fontenot et al. 2016; Gray Brunton et al. 2014; Jaiswal et al. 2020; Martin et al. 2011; Miller-Day et al. 2023; Wyndham-West 2016; |

*(Continued)*

**Table 4.** (Continued)

| Summarized review finding | GRADE-CERQual Assessment of confidence | Explanation of GRADE-CERQual Assessment | References |
|---|---|---|---|
| **Theme 10: Government and policy** | | | |
| [Finding 29] The government's involvement and role (or lack thereof) in HPV vaccination promotion were influential in vaccine decision making. | Moderate confidence | 5 studies reflecting minor concerns regarding methodological limitations, and minor concerns regarding adequacy. | Chen et al. 2021; Petrova et al. 2015; Basnyat & Lim 2018; Hodge et al. 2011; Ross et al. 2010; |

decision-making processes among young adults. While the individual findings are presented as distinct elements with a specific theme, we acknowledge that findings may also fit or overlap with other thematic categories. The thematic categories include:

- Individual factors: influence of personal attitudes or beliefs on decision-making

- Alternatives to vaccination: behaviors/practices believed to be alternatives to the HPV vaccine

- Knowledge and information: knowledge and information about HPV/HPV vaccine

- Sex and romantic relationships: influence of sexual status and relations on decision-making

- Parents and peers: role of parents/peers in decision-making

- Physicians: role of physicians in decision-making

- Logistics: factors associated with the process of getting vaccinated

- The vaccine: components about the HPV vaccine influential in decision-making

- Gendered perceptions and bias: influence of gendered views on promotion and uptake

- Government and policy: role of government in decision-making

Theme 1: Individual factors

**Finding 1. Missed opportunity: Some young adults believed they missed the window for vaccination because they were too old to benefit from it.** Studies reported that some young adults believed they were too old to receive the HPV vaccine. This was due to lack of clarity about the vaccine's efficacy past certain ages [65,72,108,110]: *"I heard that the HPV vaccination works on girls prior to age 15. It is too late for me to benefit from the vaccination because I am too old and have been sexually active"* [108]. Some participants also believed the vaccine could cause adverse reactions when administered after certain ages [69] or associated the vaccine as one typically administered in childhood and subsequently no longer applicable when older:

> "When I came to college, people were saying, "Oh, I got the HPV vaccine recently," or whatever. I, like, "You're 20-something years old. What do you mean you just got it?" I just thought it was one of those things that you just get when you're young, like measles and like smallpox and all those vaccines." [72]

**Finding 2. Low risk perception: Some young adults believed they were not at risk for HPV-related illnesses or that they still had time to decide on vaccination.** Personal risk perception played an influential role in vaccine decision-making, with some participants feeling they were low risk for HPV-related diseases to justify vaccination

[45,66,75,81,110]: *"I think more the reason I just didn't get [vaccinated] was just because - yeah, I guess you're kind of right, since I was thinking, oh, it's not that big of a deal for me. Maybe this isn't a vaccine I 100 percent need to get right now"* [81]. Some young adults did not perceive the urgency to need to get vaccinated or believed it would never be too late for them to get vaccinated[67,101]: *"Since you can get vaccinated up to age 26, I have 5 more years to think about it"* [67].

**Finding 3. Proactive measure: Many young adults viewed vaccination as a proactive measure to safeguard their health in the present and prevent future regret if they didn't get vaccinated**. Many young adults expressed a desire to protect themselves from cervical cancer [64,67,71,89,100,103] other diseases associated with HPV [52,71,89,106,110]: *"When I heard what the shot can help prevent- cervical cancer - I wanted to get it"* [67]. Some individuals recalled learning about HPV and the vaccine during specific life events, such as pregnancy/health diagnoses which served as the cue to action to get vaccinated [47,113]: *"I kind of panicked a little bit when I was diagnosed with HIV, but then I'm like, OK, I'm getting everything. So it's like -- I got it [HPV vaccine]"* [47]. Participants who had experienced HPV-related symptoms expressed an increased interest in vaccination [60,78]: *"I started developing symptoms (warts), I kind of had a suspicion that's what it was. I thought I wouldn't be really suited for the vaccine, but then my doctor said that it might actually help"* [60].

Concerns about long-term health and future relationships drove some individuals to prioritize HPV prevention and avoid future regret [64,106]:

"…I didn't want to eventually be married and have that, you know, worry in my mind about HPV." [106]

"I think it would be a good thing to do. Instead of sitting there and be like "ok, I'm not gonna do it, then then all of a sudden you go to the doctor and they say you have cancer, you gonna think back like, "Oh I should've continued the shots instead of doing all that." [64]

**Finding 4. Low-stakes preventive measure: Several individuals mention the simplicity of the decision to get vaccinated, seeing it as a low-risk preventive measure with potentially significant benefits** [54,64,72,75,78,89]

"Well, it doesn't take much to vaccinate people, so I guess it is helpful. Just go and get a shot and you are all set." [89]

"There is no harm in getting the vaccine. It's a preventive measure. It's like one of those there's nothing to lose type of thing." [72]

Theme 2: Alternatives to vaccination

**Finding 5. Reliance on pap smears: Young adult women believed that regular Pap testing negated the need for HPV vaccination.** Some young adult women believe regular cervical cancer screenings were sufficient to monitor their sexual health, with many questioning the necessity for vaccination if they still had to continue getting screened [65,76,88,108]: *"Cervical cancer is treat-able if caught early, so there would be no need to get a vaccine if you are doing an annual Pap smear"* (US) [65]. In one study of participants from multiple countries, female participants also express costed savings from not getting vaccinated and just getting screened:

"Screening is something that might be...even more effective, then if you keep track you're going to avoid it and all that. Had I known it earlier, we would have saved a lot of money from vaccines [laughter]. If I'm going to have my check, what's the use of it?" [88]

**Finding 6. Condom use and other precautions: Young adults expressed precautions such as condom use, abstinence, and decisions on sexual partners were sufficient to protect against HPV, rendering the vaccine unnecessary.**

Some participants believed that making selective decisions about who to have sexual relations with and using barrier methods are effective in preventing HPV infection, leading to a perception that vaccination is unnecessary [67,108]:

> "My mother said to just not be stupid about sex … I'm not an idiot about who I sleep with. I really feel like HPV only affects people who don't make smart decisions. If there are condoms out there today, use a condom. … I feel like there are other ways to prevent HPV… less serious ways than getting the vaccine." [67]

Condom use was highlighted as a primary method of protection against HPV [67,81,89,108] as exemplified by statements such as "*I feel like condoms will protect me*" [67] and "*I typically wear condoms so I'm not particularly worried about it*" [89]. Some individuals also expressed confidence in abstinence as a means of HPV prevention, questioning the need for vaccination [80,106]: "*I am abstinent, why should I get it?*" [80]

Theme 3: Knowledge and Information

**Finding 7. Many young adults expressed having insufficient information on HPV or the vaccine before deciding to get vaccinated. There was an expressed desire more information about HPV and the HPV vaccine.** Young adults expressed concerns about not knowing enough about HPV or the vaccine [58,59,65,71,72,81,96]. Some participants demonstrated a lack of information on HPV transmission [54,62,63], how the vaccine works [54,60,78], and the association between HPV and cervical cancer [64].

If I have more information about the vaccine...I would probably get it …. As the old adage says, it is better to look before you leap. If I have more information about the vaccine, I'm willing to get it, but I don't know what it is very well." [72]

Some young adults admitted that they had never heard of HPV before vaccination or receiving the HPV vaccine without understanding what HPV is or how it is transmitted and others [54,58]: "*I have no idea what it is honestly, I got the Gardasil shot...... my mom wanted me to... but I don't know anything about it*" [54].

Participants expressed a desire for more information and awareness of HPV and the vaccine [65,72], which ranged from general knowledge about the benefits of the vaccine [58,59] to more scientific/research- related information on the vaccine [66,89]: "*It'd be very unlikely (to accept the vaccine) because I'd have to-like I said earlier-I'd have to go on the Internet and just see what has other researchers come up with and other studies and other experiments on it-what would all that calculate to*" [89].

Theme 4: Sex and romantic relationships

**Finding 8. Sexual activity and behavior: Some individuals believe that vaccination against HPV is only necessary if one plans to engage in sexual activity or has many sexual partners.** Some young adults expressed a lack of perceived need for vaccination because they were not currently sexually active and therefore did not need the vaccine [54,59,63,64,71,75,101,103,106]: "*I have no sexual experience, so I do not think I am at risk of getting cervical cancer. Cervical cancer is caused by sexual intercourse. I think those who plan to have sex may find the vaccine more useful. There is no need for me to get vaccinated if I do not have sex or get married in the near future.*" [101]

Additionally, young adults also believed that people who had a lot of sex partners should get vaccinated [67,81,88,106,110]: "*So, it will make sense to do it if you are going to be out here just engaging in a lot of sex with a lot of people, it makes sense to protect your-self*" [81].

**Finding 9. Relationship status influenced risk perception and subsequent vaccination behavior among young adults.** Many young adults in committed or monogamous relationships often perceived themselves to be safer and perceived being at lower risk for HPV infection, thereby negating the need for HPV vaccination[67,78,80,81,89,106]: "*I feel like if I were in - I don't know like single or perhaps in like a non-exclusive relationship, it's something that I would think about a little bit more. But as - yeah, I'm just like - it - it hasn't really felt as relevant for me, but I definitely think that*

*there's like a lot of benefits to it"* [106]. Conversely, a few young adults expressed interest in vaccination or that they did get vaccinated because of the additional safety and protection it offered even if they were in a relationship [106,107]: *"Yes, only because like we were sexually active, and I was like any kind of like extra protection I was game for. He -he had been with other people before, and I'm like you can't really tell when somebody has HPV; he might not even know"* (P19, single and dating, vaccinated) [106]. Changes in relationship status also appeared influential. Some participants indicated they would consider getting vaccinated if they became single [81] while others expressed that they got vaccinated after their relationship status changed: *"I wasn't going to get it, but I broke up with my boyfriend. That's when I had my first shot"* [67].

**Finding 10. Parental beliefs about sex and the HPV vaccine influenced young adults' vaccine attitudes and decisions.** Participants reported feeling influenced by their parents' beliefs regarding vaccination and sexual behavior. Parental beliefs included perceptions that getting the HPV vaccine advocates having sex [58,65,67,79,100,101]; monogamous relationships negate the need for vaccination [67]; or attitudes about morality and protecting one's virginity till marriage [101]. In these cases, young adults were deterred from getting seeking vaccination.

> "My mother was very anxious after learning that I wanted to have the cervical cancer vaccine. She said only girls who want to do something bad [have sex] want to have such vaccination, and there is no need for me to do so. My mother said it is important for good girls to protect their virginity until they get married. The reaction of my mother was so strong that it made me dare not mention it again." [101]

Theme 5: Parents and peers

**Finding 11. Mothers more than fathers played an active role in HPV vaccination process among young adults.** Many participants cited their mothers as significant sources of influence in getting the vaccine, through providing information and advice, conducting research before vaccination, and advocating for their children to receive it [48,54,59,60,64,67,72,75,102]. Mothers also facilitated important cues to action such as scheduling vaccination appointments, demonstrating their active involvement in the vaccination process [67,81,96]: *"Just every time I go in for a vaccine, my mom set the appointments. I just go and get the shot; get a nice little bandage and leave"* [81]. Some participants expressed that their mothers played a crucial role in advocating for HPV vaccination in situations where other family members such as fathers were more reluctant and hesitant about the necessity of vaccination for their child [54,78]: *"And my dad was just kind of furious [about the cost and need if she wasn't sexually active] and then... My mom was like, yeah... we need to get these and it was all okay"* [54].

**Finding 12. Decision-making power: There was a mix of deferred or shared the decision to get vaccinated between young adults and parents.** Some participants deferred vaccination decisions to their parents, particularly their mothers, trusting their understanding of healthcare information and recommendations [54,64,81]: *"Anything else again any new vaccines or any diseases going around I actually leave that with my mom. Cause she will better understand what people are saying than what I would."* (Male 18, non-initiator) [64]. In other instances, some young adults engaged in discussion and shared decision-making with their parents when considering vaccination [82,103]: *"I've discussed it with my parents but we all agreed we wanted to wait a couple more years to see what the side effects were"* [103].

**Finding 13. Peer influence: Friends and acquaintances played a significant role in shaping vaccination decisions.** Young adults who believed their peers were vaccinated or had friend groups of people who were vaccinated expressed favorable vaccine attitudes and willingness to get vaccinated [54,63,67,96,107]: *"My mom didn't want me to get it done but I thought everyone else is getting it done so I might as well"* [96]. Friends were also seen as a trust source of vaccine information which influenced uptake, and, in some instances, friends would decide to get vaccinated together [48,54]: *"One day I was just talking in the group chat, and I [asked] my friend "Oh, who hasn't gotten cervical cancer jab? I'm thinking of getting it." One of my friends said that she hasn't gotten the jab as well, and then she also expressed her interest*

*of getting it. So I said, "Why not we just go together?" She did all the research about the vaccine and then I just followed along"* [48].

Conversely, some young adults described instances where negative reactions or perceived negative perceptions from their friends deterred them from getting vaccinated [101,110]: *"Some of the friends that I do have in particular would probably think that I'm just being extremely promiscuous sleeping around with guys like every night."* [110]

**Finding 14. Health history of family and friends was a motivating factor in the decision to get vaccinated.** Concerns about cancer given family history or observing others battled HPV-related illnesses emerged as a motivator for vaccination among some participants [52,63,89,103,113]:

> "Having seen my friend going through having HPV I think it's a pretty terrible thing to go through and it affects who she chooses as a future partner, she's a single woman, right. So it's going to affect how her relationships end up unravelling and that sort of thing and whom she meets. I mean a world without HPV would be nice. But I guess my own concern about my sexual health outweighed any of the potential problems that might come with it." [113]

Theme 6: Physicians

**Finding 15. Doctor recommendation: Physician recommendations (positive or negative) were influential in vaccination uptake.** Many young adults cited their doctor's positive recommendation as a significant factor in their decision to get vaccinated, viewing it as essential for their health and trusting their expertise [47,51,54,60,64,67,69,71,81,89,96,106,113]: *"It was the doctor's recommendation. I honestly wouldn't have thought about it had he not recommended it"* [60]. Participants expressed trust and confidence in their doctors' recommendations, seeing them as experts whose advice should be followed for health-related matters [48,71,89,102,107,110].

Conversely, some individuals reported their doctor either advising against vaccination [53,54,69], or never mentioned or offered the vaccine [67,69,71,79]:

> "I didn't get it [the HPV vaccine]... because my doctor told me it wasn't necessary if I wasn't sexually active or anything." [54]

> "I talked to my doctor, and he honestly told me it was basically that...if people have not been getting the shots and not been getting cancer for a long time and it doesn't run in my family, I don't have a high probability of having it." [53]

**Finding 16. Communication gap with doctors: Young adults desired better communication and more informed conversations with their doctors.** Some participants felt that doctors were reluctant to provide detailed information unless prompted, leading to uncertainty and frustration about where to find trustworthy information [69,82,88,101]:

> I : So when you did bring [HPV] up to your doctor, did she talk to you at all about the virus?

> P: No, not at all...She was like "I guess you're sexually active, like you can get it if you want it."...I think this was before I was sexually active, because this was like my annual physical freshman year of college, and I brought it up and I'm like "This is something I read I needed," and she didn't talk about side effects, she didn't talk about what can come with it, it was like me advocating for myself and she was like "You can get it if you want but like you're gonna have to pay out of pocket." [69]

Participants expressed a need for open, non-judgmental communication with healthcare providers, where they could ask questions, receive thorough explanations and comprehensive information about the vaccine, be treated respectfully and not feel rushed [47,58,59,89]:

> "The first time I was offered the HPV vaccine, I just said no, because I didn't understand what the doctor said. Some of the terms I didn't understand, and she just spoke brief about it. I was shy at the time, too shy to ask any questions.

I saw her [provider] in a hurry, and I was like, "Oh yeah, I understand," but in reality, I didn't. I just said that because she seemed in a hurry." [59]

Participants also stressed the importance of providers creating a safe and non-judgmental environment particularly given stigma and discomfort around discussing sexual health [58,89]: *"There is still some taboo feeling with discussing sex with anybody so I think it's really important that the doctor listen and be understanding of any responses that the patient might give."* [89]. This was particularly important for LGBTQ+ participants who highlighted the importance of LGBTQ+ Inclusive healthcare and preferences of receiving care from LGBTQ+ friendly healthcare providers [47,110]: *"A gay provider would be more into or up-to-date with newer things that are coming out. Especially like with the threats that are more for the gay lifestyle. Because I really don't think that my health provider would know about HPV"* [110].

Theme 7: Logistics

**Finding 17. Appointments: Scheduling and appointment challenges impacted young adults' ability to get vaccinated.**
Participants described difficulties in completing the vaccine series due to busy schedules and other responsibilities which made it a challenge finding time to go get vaccinated [47,63,64,69,82,102,103,105]. Young adults expressed interest in combining HPV vaccination with other medical visits such as annual physicals to streamline the process and make it more convenient [58,69,89,105]: *"It would be hard for me to schedule an appointment to complete the HPV vaccine series. It would be easy to get the HPV vaccine if offered during my clinic visits"* [105]. For some participants, HPV vaccination was simply not a priority to make an effort for [69,103]: *"I never got around to it and to be honest I don't think it's a big deal to change my schedule for"* [103].

**Finding 18. Multiple shots: Some young adults found the process of going back multiple times for the three doses burdensome, leading to forgetfulness or avoidance.** Challenges with appointments were compounded by the need for multiple visits to complete the HPV vaccine series which young adults found inconvenient [47 67,69,75,81,105]:

"It's so troublesome…it's not like maybe 3 times a week you can get it over and done with. It's over more than a year, it's really long…" [75]

"I would have gotten it… it's just that getting three shots across nine months is a hassle, and I've been moving around a lot in the last couple of years" [67]

Some participants were unclear about the number of doses required for the HPV vaccine and what to do if they missed doses, leading to missed opportunities for completion [63,96]: *"didn't even know there were three shots involved with it until I came in"* [63].

**Finding 19. Accessibility: Transportation and access challenges hindered HPV vaccination uptake.** Participants cited lack of transportation or long distances as barriers to HPV vaccination [55,64,65,67,82]. Alternative locations such as pharmacies, emergency rooms or vaccine drives in atypical settings were also discussed as preferred venues to receive the HPV vaccine and increase accessibility to get vaccinated [64,82,96,110]. For some participants on college campuses, research authors noted there seemed to be an unawareness of the vaccine availability at the local campus clinic [64,75].

**Finding 20. Reminders: Reminders were cited as helpful in ensuring adherence to the vaccination schedule.**
Whether from providers or parents, or personal calendar systems to track information, young adults relied on reminders to aid them remember appointments and get vaccinated [47,64,96]. Several individuals mention that their mothers were responsible for scheduling their vaccine appointments, indicating a dependency on family members for healthcare management [67,81,96]: *"It was up to my mom to book the appointment -again"* [96].

**Finding 21. Mandates: Young adults had mixed views on mandates for HPV vaccination.** Some participants opposed mandates for HPV vaccination [71,92]. Other participants expressed compliance with mandates as part of requirements for school enrollment [64,81,100], support of public safety efforts [89], or trust in the government making vaccination compulsory [75].

Theme 8: The vaccine

**Finding 22. Vaccine cost: Young adults expressed concerns about paying for the HPV vaccine, citing high out-of-pocket costs and need for government subsidies.** Participants expressed difficulties in paying the cost for the vaccine series and subsequently opting out of vaccination [51,52,62,64,67,75,81,96,101,105,108]. Young adults also highlighted the impact of insurance coverage on their ability to afford the vaccine and decision to get vaccinated [53,63,65,89,108,110].

> ""The vaccine is too expensive. It costs almost my whole monthly wages from my part-time job. If I really have the need, I would not mind spending my wages on the vaccination. However, I am still unmarried so I do not think I have the need at this moment. Spending such a large amount of money on a vaccine that is not useful for me at this moment is not worthwhile." [101]

> I would have received the HPV vaccination if my health insurance covered it at that time or if the government paid for the HPV vaccination." [108]

Parental willingness or refusal to pay was also influential in the decision to get vaccinated [54,67,75,100,101]:

> "My mother said there is no use for me to get vaccinated except if I want to do something inappropriate [having sex]. As my mother discourages me to get vaccinated, I will not have it because she will not pay for it or sponsor me. I just have one part-time job but the wages are insufficient to pay for the vaccination." [101]

> Don't know [how much I would be willing to pay for the vaccine]. Depends on how much parents will be willing to spend!" [100]

Young adults called for governments to subsidize the cost of the vaccine, suggesting it would encourage more people to get vaccinated [51,52,75,96]: *"The main concern is the price. If the government provides more subsidies to make the vaccine cheaper, maybe (reducing) several hundred or hundred and something, it may make people more willing to be vaccinated"* [51].

**Finding 23. Side effects: Young adults expressed concerned about the side effects of the HPV vaccine.** Across many studies, participants expressed worries about potential side effects of vaccines. Participants were concerned about not knowing the long-term effects of the vaccine or how the vaccine will affect them [51,59,65,76]. Young adults were also concerned about the (unknown) ingredients of the vaccine and not wanting them in their body [59,64,71,72]. Concerns about the vaccines side effects were also fueled by the negative stories and experiences others had of getting vaccinated [54,65,69,88,101,103].

> "I trust my body. To be honest, I don't feel comfortable having it [HPV vaccine] in my body. I feel like I don't need it. I'm not even active [sexually] like that. I don't want to put something in my body if I don't know how it's going to affect me. I'd rather not risk it. I'm actually kind of scared to even do it" [59]

> "Only one or two friends of mine have had the cervical cancer vaccine; they commented the shot was particularly more painful than other vaccines. Also, one of my friends suffered from fever, dizziness, extreme tiredness and a rash for some days just after the first dose, and she dared not continue the remaining course, so I do not think about receiving the vaccine at this moment." [101]

**Finding 24. Understanding of vaccine effectiveness: Young adults questioned the effectiveness of the HPV vaccine given the limited strains it covers and lack of 100% protection.** Doubts about the effectiveness of vaccines were expressed particularly given the vaccine does not guaranteeing 100% protection [62,76,106,110,115]: *"Um, of course, now*

*it's [risk for HPV] definitely much, much lower. There's still a risk because the vaccine doesn't work 100 percent,"* [106]. In one study where participants had also received cytology testing, individuals with abnormal cytology results who also received the vaccine expressed frustration with the vaccine: *"The vaccine was a waste of time. I would say it's pointless and it's not doing what it's supposed to"* [115].

Some young adults also questioned the usefulness of vaccines if they were already sexually active or have had multiple partners, which they felt potentially negated the vaccine's effectiveness [76,88,115]: *"Well, we are already sexually active, and you are supposed to get the vaccine before you become active, because then it is most....I forgot the word... most effective. Actually, at this point, you can't be sure about the effectiveness of the vaccine"* [88].

Additionally, the knowledge that the HPV vaccine only protects against specific strains of HPV led some young adults to believe this undermined the effectiveness of the vaccine[54,65,96]:

"You've got to take into consideration that vaccination is only preventative for certain kinds of HPV.'' [65]

"... because like what's going on with swine flu right now and how they will vaccine like one part of the disease... but then there'll be another outbreak of a different [type] so I kind of think about that with the HPV; like what if they vaccinate me and I still get cervical cancer, like a different type of cervical cancer so that why I'm leery…" [54]

**Finding 25. Development of the vaccine: Concerns were raised about the vaccine's novelty, hastened development, and lack of long-term studies.** Some participants believed the vaccine development was rushed [54,88,113] and many others cited the absence of long-term studies on the vaccine's effects, and subsequent wariness about being early adopters or "guinea pigs" for the vaccine [54,62,64,67,115]. Young adults also expressed concerns about the vaccine's safety and efficacy, emphasizing the need for longer investigation to determine potential side effects and effectiveness [51,66,67,75,88,108,113].

"against it because it just wasn't very tested; it seemed to me like they approved it before they even were finished testing it in everybody and I thought that was really jumping the gun and not very safe […] the regular vaccines like that you get through public school and... ones that have been around for a long time, I'm not that worried about. But since this is so new... that makes me kind of nervous." [54]

"... I know that this vaccine has not been introduced for long, only around 10 years. Therefore, I think it requires long time observation to determine the effect." [51]

Inaccurate knowledge about how vaccines work also fueled some of the wariness and skepticism [54,115]:

"[T]he same concerns I have with any vaccine is... you're getting the virus, it's a very weak virus but you're still getting the virus and you're taking the risk that... maybe if the vaccine that you're getting is not as weak, like the virus as weak as it should be, then maybe you're more susceptible to actually like breaking out and getting the STD. But also just HPV, it seemed really hastened... When it came out for everyone to get it and it didn't seem like it was very tested to me." [54]

In one study involving UK participants, the change to quadrivalent vaccine led some participants who received the bivalent vaccine to believe they may not be fully protected because the bivalent vaccine was not as effective: *" It's different now. It's been researched and changed. [...]I hope that other girls aren't thinking 'Well I'm fine because I've got it [vaccination]"* [115].

**Finding 26. Pharmaceutical companies and business motive: Young adults expressed skepticism towards pharmaceutical companies and their motives, suggesting that public panic and greed may influence vaccination**

**campaigns.** Participants questioned the motive of vaccination promotion and who stood to most benefit [76,101]. Young adults also criticized the advertising tactics used for the vaccine suggesting campaigns may be manipulative or misleading by not adequately informing people about potential risks and benefits [62,88]; they also questioned *who* (e.g., drug companies vs the government) should be promoting the vaccine [51,76] and were critical of the marketing of the vaccine as a product [88,101].

"Well nowadays, whatever they offer me a vaccination against, a medicine, or whatever it is, and they tell me it is very good, I will always keep in mind who will win the most. Is it my health, is it this guy's pocket, the government or state, or whoever it is, does not matter" [101]

"I think it [marketing campaign] was quite directed...which then to me I think that's wrong, you shouldn't be advertising, you know thinking about the audience like that because it's not a product, it's er.. medical, you're not trying to sell this!" [88]

"... the government instead of the drug companies should take the role to promote (the vaccine)...If the drug companies take the initial role to promote, it could only promote through limited media sources... But if this is the government to take the role, the vaccine can be introduced in sex education." [51]

Theme 9: Gendered Perceptions and Bias
**Finding 27. Gendered perceptions of HPV risk: Many participants exhibit misconceptions about HPV being primarily a women's disease.** Some young adult men believed HPV does not affect men and that only women needed to be vaccinated because HPV affected them the most [45,60,66,69,80]. In some instances, men expressed knowing they could be carriers of HPV and pass it on to their (female) partners without themselves getting sick [58,60,69].

"I think the vaccine is for woman... I heard it's only, the vaccine...no no, the virus only target woman, even if the man got it, it wouldn't affect them. But he would be the carrier of that [virus], he might transmit that [virus] to his partner who is a female...." [69]

One study also demonstrated they gendered perception was not solely a male issue; a female participant also expressed no knowing men could get vaccinated [72]:

"I really didn't know that men could get even the vaccine. My friend and I were talking about it. He's like, "Yeah, I had Gardasil, too," and I was like, "Really? You don't have cervix. You're a guy." He was like, "It's a vaccine for HPV." I was like, "What?" It didn't make sense to me at that time because I thought this vaccine is for women only."

**Finding 28. Gender biased promotion: The gendered marketing and promotion of HPV vaccines led to perceived unfairness in who gets access to vaccine and who is responsible for getting the vaccinated in the context of sexual health.** Some young adults criticize the gendered nature of vaccination campaigns that were predominantly aimed at women, questioned the fairness of offering the vaccine primarily to girls and women and advocated for equal access to vaccination for all genders/sexes [58,62,81,113]:

"I'm really skeptical about the vaccine, particularly because it's mass marketed towards girls and women and absolutely no focus on boys and men. And you know it's a vaccine for HPV, it's not a vaccine for cervical cancer even though the strains it targets can lead to cervical cancer. But I just feel that if we actually want to reduce cancer we probably should be vaccinating everyone. I think it is interesting that they're targeting only girls and women because there hasn't been any long-term research so we don't really know what's going to happen ten years down the line, 25 years down the line." [113]

"I think the biggest thing is just letting [boys and men] know it's an option because I, obviously, when I was 18 they didn't tell me, so I didn't get it. But the second I was told about it, I did. So, I think it's just a matter of being aware that it exists." [81]

Men and transgender individuals particularly noted the lack of representation and targeted advertising towards their demographic; this led to them feeling less informed about the risks and importance of vaccination [47,58,66,69]:

"It was that commercial. And I it stands out to me so much because I remember "HPV can lead to cervical cancer." And I just remember hearing it and it stuck in my brain forever, and I just remember being like "Thank god I don't have ovaries." (I: Mhm.) But then I googled it one day cause I got, I was like "What if I get it one day" and I was like, "wait it can affect me too?" [69]

"The biggest thing, again, was primarily marketed towards women, which - cisgender women - is, then, the question of, well, then, why did I bother, as a male? You know, as a cisgender male, it doesn't make sense for me." [47]

"There is a misconception that if you're a gay man, you don't need to get it [HPV vaccine]." [58]

A couple of studies revealed a discussion on the unequal distribution of responsibility for sexual health between genders, with some feeling that HPV vaccination campaigns unfairly target women and reinforce gender biases. Young adult women raised concerns about the stigmatization of women's sexuality and the unequal burden of responsibility placed on them in sexual relationships, being a "scapegoat" of men's sexual risk-taking behavior [77,113]:

"Ita: It sends a message that it's a girl's responsibility. It's a girl's responsibility to be on the pill and its just perpetuating that. When I was a teenager it was always the girls responsibility to carry a condom, it wasn't for boys to worry about. Boys need to be responsible for their sexual health too.

Anna: Yeah, straight men may get reassurance from the fact that their partner has had the vaccine so they cannot catch it." [77]

"Okay, so I have a problem with the vaccine even though I got it. The way it's being marketed towards girls stigmatizes girls' and women's sexualities as being the source of the problem. Just like when you look at, on having responsible protective sex, again a lot of the onus is put on girls to be sort of the arbitrators of sexuality so that they have to monitor their partner's behaviour and make sure to take on that responsibility of being responsible for both her and her partner without attributing very much responsibility to the man who is involved in the sexual relationship." [113]

Theme 10: Government and policy
**Finding 29. The government's involvement and role (or lack thereof) in HPV vaccination promotion were influential in vaccine decision-making.** For some young adults, the government's involvement in and approval of the HPV vaccine was viewed as confirmation of its importance [48,96]:

"Well that kind of reassures you as well. If the government is making sure you have those jabs and they're paying £400 for you to have them, then it does work and it helps people…" [96]

"For me, if the government thinks that it's important enough for you to use your CPF [Central Provident Fund] money for it or even Medisave money for it then it must be important enough. Enough important [to be] on the radar for them. I mean you won't see them [government] giving flu vaccination and tapping from Medisave right?" [48]

Conversely, lack of government involvement in HPV vaccine promotion raised questions among participants [52,88]:
*"Well, for me the primary reason is that, in reality, if the government wanted to support this thing, it would have provided*

*more information, as a first step towards action"* [88]. Similarly, mistrust in the government led some participants to be critical of government initiatives: *"For Natives, vaccinations are crazy. The whole history with the IHS (Indian Health Service), I'm not sure people believe they can trust them[the government]"*' [65].

## SEM Categorization of findings

To enhance our understanding of the factors that influence young adult vaccination decision-making and uptake, we organized our review findings across the domains of the socioecological model (SEM). The SEM proposes an interplay and interdependence of factors within and across different ecological levels that influence health behaviors [117]. We acknowledge that not all findings neatly fit into the constructs of the SEM and that some findings may span multiple domains–we placed findings in ecological levels that was most representative of their level of influence as presented by the data (Fig 2).

## Discussion

This systematic review aimed to qualitatively synthesize the literature on young adults' perspective on HPV vaccine decision-making and the associated barriers and facilitators that influence their vaccination uptake. The current review

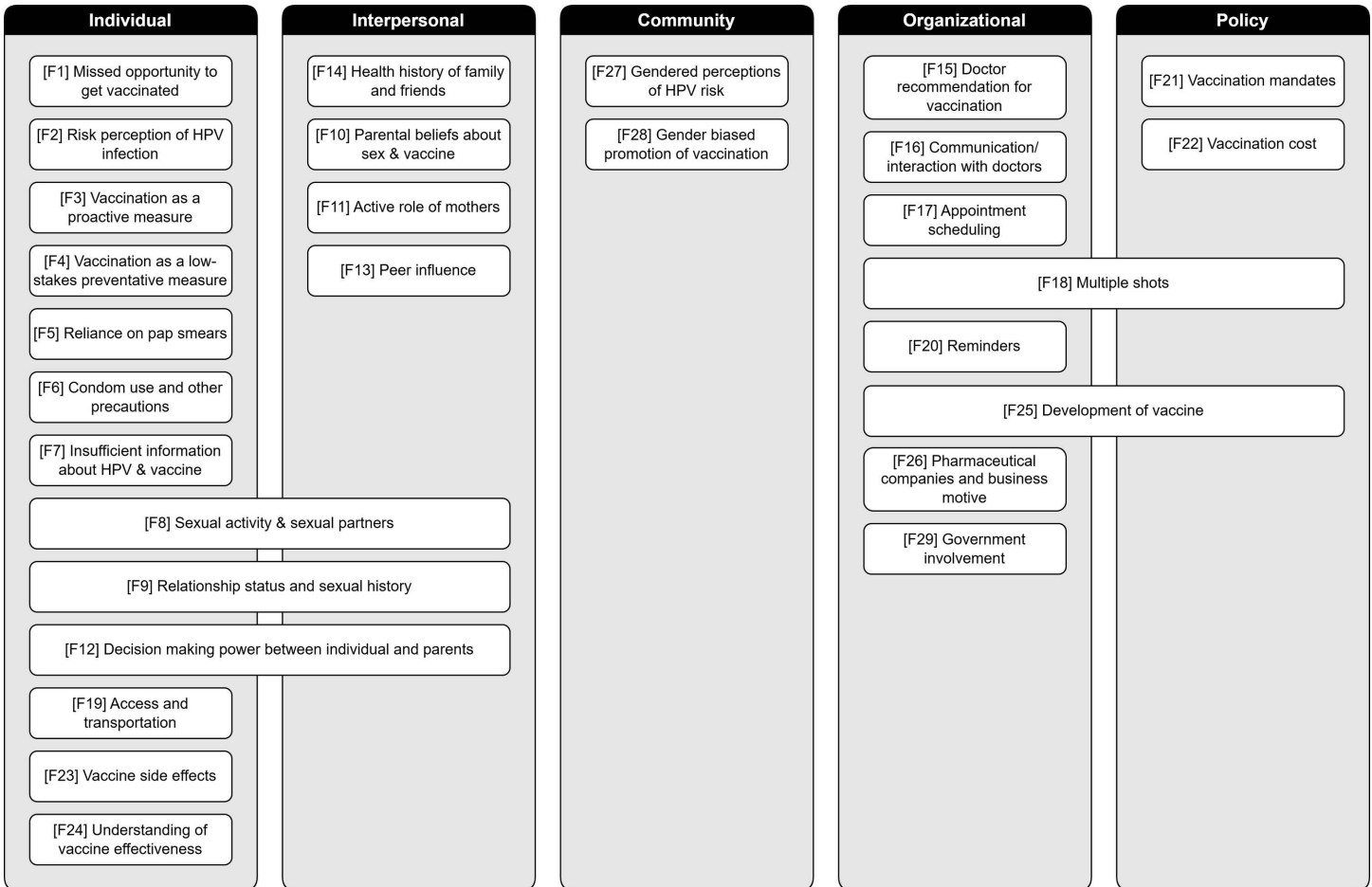

**Fig 2. Socioecological characterization of review findings.**

identified 68 studies from 71 articles that met inclusion criteria; of these, 42 studies exclusively sampled young adults between the ages of 18 and 26 years which formed the basis of the results. We identified 29 findings across 10 thematic categories which revealed a complexity of factors that are influential in decision-making and vaccination uptake among young adults. We found that some young adults believed that they had aged out of eligibility for HPV vaccination or that their sexual history would hinder the effectiveness of the vaccine and precluded them from the vaccine's protective benefits. We also observed that some young adults considered condom use and regular screenings as alternatives to vaccination in preventing HPV infections. Lastly, we observed that young adults more frequently spoke of their mothers as playing an influential and active role in the vaccine decision-making and uptake process.

The studies that contributed to this review spanned multiple countries, representing various geographic, political, and economic contexts. The policies and guidelines adopted by different nations may influence and impact vaccination uptake. For example, the World Health Organization (WHO) recommends one or two HPV vaccine doses for girls and women 18–20 years old and 2 doses for women older than 21 years [8]; in contrast, the CDC in the US recommends 3 doses of the HPV vaccine for all individuals aged 18–26 [118]. The observed differences in the two policies of who is advised to get vaccinated, and the number of doses required, have implications on vaccination uptake as demonstrated by our findings. On the issue the vaccine schedule, our review findings showed that returning for multiple doses was a barrier for some young adults. On the matter of persons to get vaccinated, the CDC policy to vaccinate everyone regardless of gender/sex aligns with our findings advocating for gender balance in the marketing and promotion of the HPV vaccine and the responsibility of everyone – not just females – to get vaccinated. This in no way diminishes the vital importance of WHO's position that has a primary focus on cervical cancer prevention [8] and optimizing vaccination in under-resourced countries who experience a disproportionate burden of cervical cancer[119]. However, given this review's findings, we encourage that where feasible, the promotion of HPV vaccination be inclusive and gender neutral.

Findings from this review highlight several constraints young adults experience in getting vaccinated, including cost of and accessibility to the HPV vaccine; specifically, some participants cited lack of transportation access or long travel distances as a barrier to getting vaccinated, and others expressed difficulty in paying for the vaccine series due to lack of insurance, high out-of-pocket expense, or lack of government subsidy. Globally, 143 (75%) countries include the HPV vaccine in their national immunization programs (NIP), providing the vaccine at no or reduced cost [120]. One study conducted on NIPs in Asia-Pacific region found that only 3 of the 10 countries evaluated include catch-up vaccination in their immunization program –and notably no country included males in their NIP [121]. Further investigation should be undertaken on the eligibility and inclusion of HPV catchup vaccinations included in NIPs globally. Additionally, many primary HPV immunization programs offer vaccine delivery in school settings and these school-based programs have shown significant improvements in adolescent vaccine coverage [9,122]. For young adults, college campuses are posited as the ideal setting to address HPV vaccination disparities [123]. One challenge with college settings is that while HPV vaccines may be available at college health centers, the utilization rate of campus health services for vaccination may not be fully maximized. A national study conducted in the US found that only 9% of visits to campus health centers at 4-year institutions involved a vaccination of any type [124]. Future research should investigate the utilization of campus health services for HPV vaccination and additionally, evaluate the processes by which unvaccinated students are nudged/encouraged to get vaccinated. An assessment of vaccine policies across 96 higher education institutions in the US found that no institution required the HPV vaccine for school entry and only 36.5% of institutions recommended it for enrollment [125]. Given the prevalence and associated cancer burden of HPV infections, it is imperative that all avenues be pursued to increase the HPV vaccine coverage among young adults.

While young adults' HPV vaccination decision is generally thought of to be autonomous and separate from parental attitudes, involvement, or supervision [123,126], our review findings suggest that parents may still be influential in the process—whether it was providing information, scheduling appointments, or being deferred to for decision-making.

Additionally, our findings also contextualize the evidence supporting the importance and implications of physician recommendations for HPV vaccination; young adults who were willing to get the vaccine were met with either support or reluctance by physicians in providing information about the vaccine or advocating for its necessity. It also became evident from our findings that the nature of the interaction with physicians may play an important role; the "rushed" nature of medical appointments may discourage openness and dialogue that some young adults are seeking when deciding on vaccination. Parents and physicians are some of the key trusted sources of medical information for young adults [127,128] which may account for the importance of these entities in vaccine decision-making. Future interventions on catch-up vaccination should consider engaging vaccine-hesitant parents and physicians in some capacity, given the influential roles they play in the decision-making process.

There are several limitations to account for with this review. During the screening process, we encountered studies that also included participants under 18 years or over 26 years, and excluded articles where we were unable to extract data explicitly from participants in our age group of interest; consequently, literature providing important information on the topic may have been excluded from this review. Similarly, given the explicit focus on young adults aged 18–26, these findings may not be generalizable to other age groups. Additionally, given we purposively sampled articles to manage the volume of the qualitative data for analysis, we may have also excluded rich data to contribute to the review findings. It is also worth noting that the age cut of 18 years used for this review is based on the legal age of consent standard in the US; given we did not restrict the geography of studies during our search, it is possible that studies from other countries sampling young adults of legal consenting age younger than 18 years may have been excluded. Lastly, while this review included studies across 14 countries, we did not present the findings in a country or regions specific manner, which may hinder generalizability across geographic locations given differences in norms, culture and policies that may exist from country to country.

Despite these limitations, this review contributes valuable insights on HPV vaccination among young adults. The novelty of our study as one of the principal QES on catch-up HPV vaccination, presents findings that have been systematically evaluated and thus offer greater assurance in the review findings. Our findings also underscore the complexity of factors at play across multiple ecological levels, any combination of which may aid or impede vaccination uptake among young adults. There continues to be a need to provide accurate and reliable information about HPV and the vaccine; similarly, while efforts have been made by research and medical communities to focus messaging about the HPV vaccine on cancer prevention rather than sexual transmission [129], our findings illustrate a need for some education around sexual health and HPV that emphasizes the importance of HPV vaccination regardless of sexual history or sexual health behaviors (e.g., condom use or monogamous relationship status). Furthermore, our findings on parental influence on vaccine decision-making underscore the need for continued research assessing the impact of parental influence on older adolescents' health behaviors; investigations should also explore social network influences of vaccine attitudes/beliefs on vaccination outcomes, particularly within a familial network. Lastly, efforts should also continue of help keep the cost of the HPV vaccine affordable, particularly through policy interventions that expand/increase insurance coverage for young adults or promote the adoption of national immunization plans that include young adults.

## Conclusion

HPV vaccines remain the most effective form of prevention against HPV infections and HPV-associated cancers. The findings from this review present important considerations to account for when designing and implementing interventions, programs, and policies aimed at addressing HPV vaccination disparities among young adults.

## Supporting information

**S1 Appendix. Database Search Terms.**
(DOCX)

**S2 Appendix. Identification of New References following Search Strategy Updates**.
(DOCX)

**S3 Appendix. GRADE CERQUAL Evidence Table**.
(DOCX)

**S4 Appendix. Prisma Checklist**.
(DOCX)

**S5 Appendix. Excluded Studies**.
(PDF)

## Author contributions

**Conceptualization:** Namoonga M. Mantina, Priscilla Anne Magrath, Deborah Jean McClelland, Leila Barraza, John Ruiz, Purnima Madhivanan.

**Data curation:** Namoonga M. Mantina, Jonathan Smith, Flavia Nakayima Miiro.

**Formal analysis:** Namoonga M. Mantina, Jonathan Smith, Flavia Nakayima Miiro.

**Methodology:** Namoonga M. Mantina, Priscilla Anne Magrath, Deborah Jean McClelland, Purnima Madhivanan.

**Writing – original draft:** Namoonga M. Mantina.

**Writing – review & editing:** Namoonga M. Mantina, Jonathan Smith, Flavia Nakayima Miiro, Priscilla Anne Magrath, Deborah Jean McClelland, Leila Barraza, John Ruiz, Purnima Madhivanan.

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
