## [Decision Letter · Decision Letter 0]

1 Jan 2025

PONE-D-24-44380Perspectives of HPV Vaccine Decision Making Among Young Adults: A Qualitative Systematic Review and Evidence SynthesisPLOS ONE

Dear Dr. Mantina,

Thank you for submitting your manuscript to PLOS ONE. After careful consideration, we feel that it has merit but does not fully meet PLOS ONE’s publication criteria as it currently stands. Therefore, we invite you to submit a revised version of the manuscript that addresses the points raised during the review process.

Your manuscript was reviewed by two experts in the field. Both identified many important problems in your submission. Please review the attached comments and provide point-by-point responses.

We look forward to receiving your revised manuscript.

Kind regards,

Yury E Khudyakov, PhD

Academic Editor

PLOS ONE

Journal Requirements:

2. As required by our policy on Data Availability, please ensure your manuscript or supplementary information includes the following: 

Reviewers' comments:

Reviewer's Responses to Questions

**Comments to the Author**

1. Is the manuscript technically sound, and do the data support the conclusions?

Reviewer #1: Yes

Reviewer #2: Yes

2. Has the statistical analysis been performed appropriately and rigorously? 

Reviewer #1: Yes

Reviewer #2: N/A

3. Have the authors made all data underlying the findings in their manuscript fully available?

Reviewer #1: Yes

Reviewer #2: Yes

4. Is the manuscript presented in an intelligible fashion and written in standard English?

Reviewer #1: Yes

Reviewer #2: Yes

5. Review Comments to the Author

Reviewer #1: Background:

• The paragraph discussing the need for qualitative research could be streamlined to enhance cohesion.

• To improve readability, authors should consider combining points on the CDC and WHO vaccination recommendations. Then proceed to the discussion on disparities in vaccination rates and their implications on public health.

• Authors should add a brief concluding sentence summarizing the gaps in research and the importance of targeted interventions for HPV vaccination in young adults. This will provide a stronger ending.

Methods:

• The authors should consider introducing subheadings for each subsection (e.g., "Eligibility Criteria," "Information Sources and Search Strategy," etc.) to improve navigation for readers.

• Authors should explain briefly why the review focuses on young adults aged 18-26 specifically. Adding a sentence to contextualize this age range would enhance the reader's understanding of the focus.

• In the sentence, “In alignment with the approaches taken by other qualitative review authors [30], we purposively sampled a subset of the studies for analysis…,” briefly clarify why purposive sampling was used (e.g., to enhance the depth of thematic analysis) as this adds rationale to your methods.

• In Data Synthesis, this section could be expanded slightly to explain the coding process more clearly, particularly how codes were created, discussed, and agreed upon. The authors should explicitly mention how discrepancies in coding were handled to enhance transparency.

• The authors should add a sentence to explain why GRADE-CERQual was chosen (does it assess confidence in qualitative evidence rigorously?). This would give this section more depth.

• Was there ethical considerations? If yes, consider including a statement on ethical considerations or any limitations of the methodology, particularly around sampling and data extraction.

• Ensure consistent formatting for tool names and software (e.g., "DistillerSR," "EndNote," "REDCap").

• Use italics for journal names, e.g., BMJ Open.

• Ensure that all acronyms, like "QES" and "CASP," are spelled out on the first mention to improve readability.

Discussion:

• When discussing barriers like “the cost of and accessibility to the HPV vaccine,” please provide more context or examples to illustrate these barriers. For instance, “limited financial support? etc”

• While the authors discuss parental and physician influence, they should consider addressing why these factors are relevant for young adults specifically. A sentence explaining why young adults may still be influenced by their parents or need strong physician support would add depth.

• The authors should expand on the limitations section, particularly regarding the geographic and age restrictions. They could clarify how these limitations might affect the generalizability of their findings.

• The authors should also consider suggesting future research directions based on these limitations, such as studying younger adolescents' transition into adulthood or assessing the impact of parental influence on older adolescents.

Reviewer #2: Perspectives of HPV Vaccine Decision Making Among Young Adults: A Qualitative

Systematic Review and Evidence Synthesis

The article is a systematic review that looks into factors that affect how young adults view HPV vaccination across the world through a two-step data extraction process. Each section of the article is well-detailed and the findings are comprehensive. The discussion offers a sufficient summary of the findings.

The manuscript is well-written is easy to follow. Here are my comments:

The ABSTRACT section is succinct and sufficient.

The research objectives can be incorporated into the INTRODUCTION section.

Table 2 [page 27] is a very good summary of the 71 articles representing 68 studies that met the eligibility criteria and the 45 articles representing 42 studies that were included and synthesized for the review.

The study is very thorough and detailed. While this is essential to the comprehensiveness of the manuscript, it may impact on the length of the paper.

There are 10 thematic categories identified [page 35]. Perhaps a one-line definition of each theme may be necessary given that not all of the themes are self-explanatory. Further, looking at the specific findings per theme, there seems to be some overlap. For instance, the findings of theme 4 (sex and romantic relationships) overlap with or may be related to the findings of theme 1 (individual factors).

If the manuscript needs to be shortened or made more concise, perhaps the themes can be combined to lessen their number and to highlight possible points of intervention, which may also enrich the discussion. For example, themes 1, 3, 4, and 9 can be placed under one theme relating to proper information; themes 5 and 6 can be under the theme regarding influences to decision-making; and themes 2, 7, 8, and 10 relate to the vaccine itself, including alternatives, logistics, and policy. The number of findings will not change, only the themes that cover them. This will make the manuscript more focused and enable it to posit more focused interventions.

The DISCUSSION section mentions that the findings “highlight several constraints that young adults experience in getting vaccinated” [page 56]. However, there is no discussion, no matter how short, on possible action areas. Some recommendation may be helpful, especially in the context of the relevance underscored in the INTRODUCTION section stated as “(T)his presents a crucial opportunity to intervene and promote catch-up HPV vaccination” [page 3].

A brief CONCLUSION section may help.

6. PLOS authors have the option to publish the peer review history of their article (what does this mean? ). If published, this will include your full peer review and any attached files.

**Do you want your identity to be public for this peer review?** For information about this choice, including consent withdrawal, please see our Privacy Policy .

Reviewer #1: No

Reviewer #2: No

---

## [Author Response · Author response to Decision Letter 1]

14 Feb 2025

File attached for response to reviewers

---

## [Decision Letter · Decision Letter 1]

7 Mar 2025

Perspectives of HPV Vaccine Decision-Making Among Young Adults: A Qualitative Systematic Review and Evidence Synthesis

PONE-D-24-44380R1

Dear Dr. Mantina,

We’re pleased to inform you that your manuscript has been judged scientifically suitable for publication and will be formally accepted for publication once it meets all outstanding technical requirements.

Kind regards,

Yury E Khudyakov, PhD

Academic Editor

PLOS ONE

Additional Editor Comments (optional):

Reviewers' comments:

Reviewer's Responses to Questions

**Comments to the Author**

1. If the authors have adequately addressed your comments raised in a previous round of review and you feel that this manuscript is now acceptable for publication, you may indicate that here to bypass the “Comments to the Author” section, enter your conflict of interest statement in the “Confidential to Editor” section, and submit your "Accept" recommendation.

Reviewer #1: All comments have been addressed

2. Is the manuscript technically sound, and do the data support the conclusions?

Reviewer #1: Yes

3. Has the statistical analysis been performed appropriately and rigorously? 

Reviewer #1: Yes

4. Have the authors made all data underlying the findings in their manuscript fully available?

Reviewer #1: Yes

5. Is the manuscript presented in an intelligible fashion and written in standard English?

Reviewer #1: Yes

6. Review Comments to the Author

Reviewer #1: Thank you for addressing most of my comments riased for this manuscript. i enjoyed reviewig your work.

7. PLOS authors have the option to publish the peer review history of their article (what does this mean? ). If published, this will include your full peer review and any attached files.

**Do you want your identity to be public for this peer review?** For information about this choice, including consent withdrawal, please see our Privacy Policy .

Reviewer #1: **Yes: ** Osmond Ekwebelem

---

## [Editor Report · Acceptance letter]

PONE-D-24-44380R1

PLOS ONE

Dear Dr. Mantina,

I'm pleased to inform you that your manuscript has been deemed suitable for publication in PLOS ONE. Congratulations! Your manuscript is now being handed over to our production team.

Kind regards,

on behalf of

Dr. Yury E Khudyakov

Academic Editor

PLOS ONE